# JAZF1 safeguards human endometrial stromal cells survival and decidualization by repressing the transcription of G0S2

Yingyu Liang[1], Siying Lai[1], Lijun Huang[1], Yulian Li[1], Shanshan Zeng[1], Shuang Zhang[1], Jingsi Chen[1], Wenbo Deng [2], Yu Liu[1], Jingying Liang[1], Pei Xu[1], Mingxing Liu[1], Zhongtang Xiong[3], Dunjin Chen [1✉], Zhaowei Tu [1✉] & Lili Du [1✉]

Decidualization of human endometrial stromal cells (hESCs) is essential for the maintenance of pregnancy, which depends on the fine-tuned regulation of hESCs survival, and its perturbation contributes to pregnancy loss. However, the underlying mechanisms responsible for functional deficits in decidua from recurrent spontaneous abortion (RSA) patients have not been elucidated. Here, we observed that JAZF1 was significantly downregulated in stromal cells from RSA decidua. JAZF1 depletion in hESCs resulted in defective decidualization and cell death through apoptosis. Further experiments uncovered G0S2 as a important driver of hESCs apoptosis and decidualization, whose transcription was repressed by JAZF1 via interaction with G0S2 activator Purβ. Moreover, the pattern of low JAZF1, high G0S2 and excessive apoptosis in decidua were consistently observed in RSA patients. Collectively, our findings demonstrate that JAZF1 governs hESCs survival and decidualization by repressing G0S2 transcription via restricting the activity of Purβ, and highlight the clinical implications of these mechanisms in the pathology of RSA.

[1] Department of Obstetrics and Gynecology, Guangdong Provincial Key Laboratory of Major Obstetric Diseases, Guangdong-Hong Kong-Macao Greater Bay Area Higher Education Joint Laboratory of Maternal-Fetal Medicine, Guangdong Engineering and Technology Research Center of Maternal-Fetal Medicine, Guangdong Provincial Clinical Research Center for Obstetrics and Gynecology, The Third Affiliated Hospital of Guangzhou Medical University, Guangzhou 510150, China. [2] Department of Obstetrics and Gynecology, Fujian Provincial Key Laboratory of Reproductive Health Research, The First Affiliated Hospital of Xiamen University, Xiamen University, Xiamen 361102, China. [3] Department of Pathology, The Third Affiliated Hospital of Guangzhou Medical University, Guangzhou 510150, China. ✉email: gzdrchen@gzhmu.edu.cn; tuzhaowei0519@hotmail.com; lilidugysy@gzhmu.edu.cn

Recurrent spontaneous abortion (RSA), also known as recurrent pregnancy loss (RPL), is a highly heterogeneous condition that is defined as the failure of two or more consecutive clinically recognized pregnancies (documented by ultrasonography or histopathologic examination)[1,2]. Beyond chromosomal errors, immune system abnormalities and uterine malformations[3], the endometrial dysfunction has also been proposed as an important cause for RSA[4].

During pregnancy establishment and maintenance, the hESCs undergo dynamic transformation during the progesterone-dominant early secretory phase, which is called decidualization[5]. The hallmark of decidualization is the differentiation of fibroblast-like hESCs into decidual stromal cells (DSCs) characterized by rounding of the nucleus, increased number of nucleoli, dilatation of the rough endoplasmic reticulum (rER) and Golgi systems[6], and secretion of growth factors and cytokines, such as prolactin (PRL) and insulin-like growth factor-binding protein 1 (IGFBP1), which are commonly used as decidualization markers. Decidualization provides the most efficient environment for early embryo development. Functional defects of decidualization lead to a variety of pregnancy complications, including RSA, recurrent implantation failure (RIF) and preeclampsia (PE)[7–9]. DSCs undergo apoptosis during decidualization in an autocrine or paracrine manner and proper apoptosis contributes to the establishment of decidualization, thereby maintaining decidua homeostasis and promoting embryo attachment and invasion[10–12]. Several models have revealed the regulatory effect of cell apoptosis on decidualization of the mouse endometrium[13–15]. Although immoderate apoptosis of DSCs has also been reported in early abortion and RSA[16–20], the underlying mechanisms of how this excessive apoptosis is caused in RSA remain obscure.

Juxtaposed with another zinc finger protein 1, JAZF1 (also known as TIP27 and ZNF802) functions as a repressor of DNA response element 1 (DR1)-dependent transcription of nuclear receptor subfamily 2, group C, member 2 (NR2C2), or testicular orphan nuclear receptor-4 (TR4)[21]. It is required for a variety of molecular functions, including anti-inflammatory[22], anti-lipogenesis[23], anti-hyperglycemia[24], transcriptional regulation[25]. Studies have shown that JAZF1 is also involved in the regulation of apoptosis and proliferation in many cells, including human spermatogonial stem cells and β-cell[25,26]. Furthermore, overexpression of JAZF1 is also implicated in downregulating phosphorylation of FKHR[27,28] (also known as forkhead box O1 (FOXO1)) and cAMP response element-binding protein (CREB)[29], which are key regulators of decidualization[30,31]. However, the function of JAZF1 in decidua homeostasis and pregnancy maintenance remains unclear.

Here, we found that the expression of JAZF1 was downregulated in the decidua tissue of RSA samples. Gene deletion assays proved JAZF1 as a key regulator of cell death via apoptosis and decidualization of immortalized hESCs or primary endometrial stromal cells (ESCs). Moreover, we identified G0/G1 switch protein 2 (G0S2) as a downstream factor of JAZF1 which mediated mitochondrial apoptotic pathway as observed in RSA decidua. JAZF1 interacted with Purβ and restrict its regulation of G0S2 transcription. Our findings proved that JAZF1 is an essential regulator of decidualization homeostasis in human endometrium, whose disruption led to excessive apoptosis and defective DSCs function during pregnancy establishment and maintenance.

## Results
### Aberrantly decreased level of JAZF1 in decidua is associated with RSA. Using our previous single-cell RNA sequencing

(scRNA-seq) datasets of decidua cells from 6 RSA and 5 matched normal first trimester decidua[7], we found that JAZF1 was significantly downregulated in all DSCs of RSA group (Fig. 1a–c, Supplementary Fig. 1a). To further validate our scRNA-seq results, we detected the mRNA and protein expression levels of JAZF1 in the first trimester decidua and primary DSCs from RSA and normal pregnancy. The information of RSA and normal pregnancy samples used for qPCR and western blot assay was in Supplementary Table S1–2. We demonstrated that both the mRNA and protein levels of JAZF1 were downregulated in RSA decidua (Fig. 1c–e). In consistent with this, we detected lower expression of JAZF1 in primary DSCs isolated from RSA decidua compared to normal group (Fig. 1f, g). The clinical characteristics of the RSA and control cases used to isolate primary DSCs were listed in Supplementary Table S3–4. Next, we performed immunohistochemistry (IHC) and immunofluorescence (IF) which also proved that JAZF1 expression was significantly decreased in the nucleus of DSCs in RSA group compared with the control (Fig. 1h–j).

To address the potential functions of JAZF1 during decidualization, we examined the mRNA and protein levels of JAZF1 in immortalized HESCs and primary ESCs during in vitro induced decidualization. As shown in Fig. 2a–e, both mRNA and protein expression levels of JAZF1 gradually increased in immortalized hESCs with the duration exposure to MPA, and cAMP cocktail, accompanied by the up-regulation of decidualization-related genes IGFPB1, PRL and FOXO1. The results from primary ESCs were consistent with this (Supplementary Fig. 1b–d). IF staining also demonstrated that JAZF1 was mainly localized in the nucleus of immortalized HESCs after decidualization (Fig. 2f). We also examined the JAZF1 expression pattern in the human endometrium by IHC staining. The results showed that the JAZF1 protein abundance was higher in the early secretory phase than the proliferative phase of the endometrium, especially in the ESCs (Fig. 2g). We also analyzed the relative expression of JAZF1 across the menstrual cycle from a public scRNA-seq data and endometrium microarray data. The results also showed that the expression of JAZF1 was upregulated in the early and middle secretory phases compared with the proliferative phase (Fig. 2h, i, Supplementary Fig. 1e). Moreover, the expression levels of JAZF1 in the proliferative and early secretory endometrium of women who suffered RSA were both lower (Fig. 2g). These results indicated that JAZF1 is a potential regulator on the establishment and development of decidualization.

### JAZF1 depletion promoted cell death via apoptosis and decidualization defect of stromal cells. Since JAZF1 deficiency extensively promotes apoptosis in β-cells[25] and DSCs undergo fine-tuned apoptosis during decidualization, we proposed that JAZF1 downregulation may lead to excessive apoptosis during hESCs decidualization. To depict its function, JAZF1 was knocked down by siRNA, knocked out using CRISPR/Cas 9 technology and overexpressed by lentivirus in immortalized HESCs before decidualization in vitro. The efficiency of knockdown, knockout or overexpression of JAZF1 confirmed by qPCR, western blotting and sanger sequencing was shown in Supplementary Fig. 2a–c. Both the knockdown and knockout approaches in immortalized HESCs and primary ESCs yielded similar results and were summarized together. As shown in Fig. 3a and Supplementary Fig. 2d, JAZF1 depletion promoted the mRNA levels of pro-apoptotic genes BAX, BAX/BCL2 and hampered the anti-apoptotic gene BCL2 under the condition of decidualization. Consistently, protein levels of BAX, BAX/BCL2, cleaved-caspase 3, cyt c were significantly upregulated

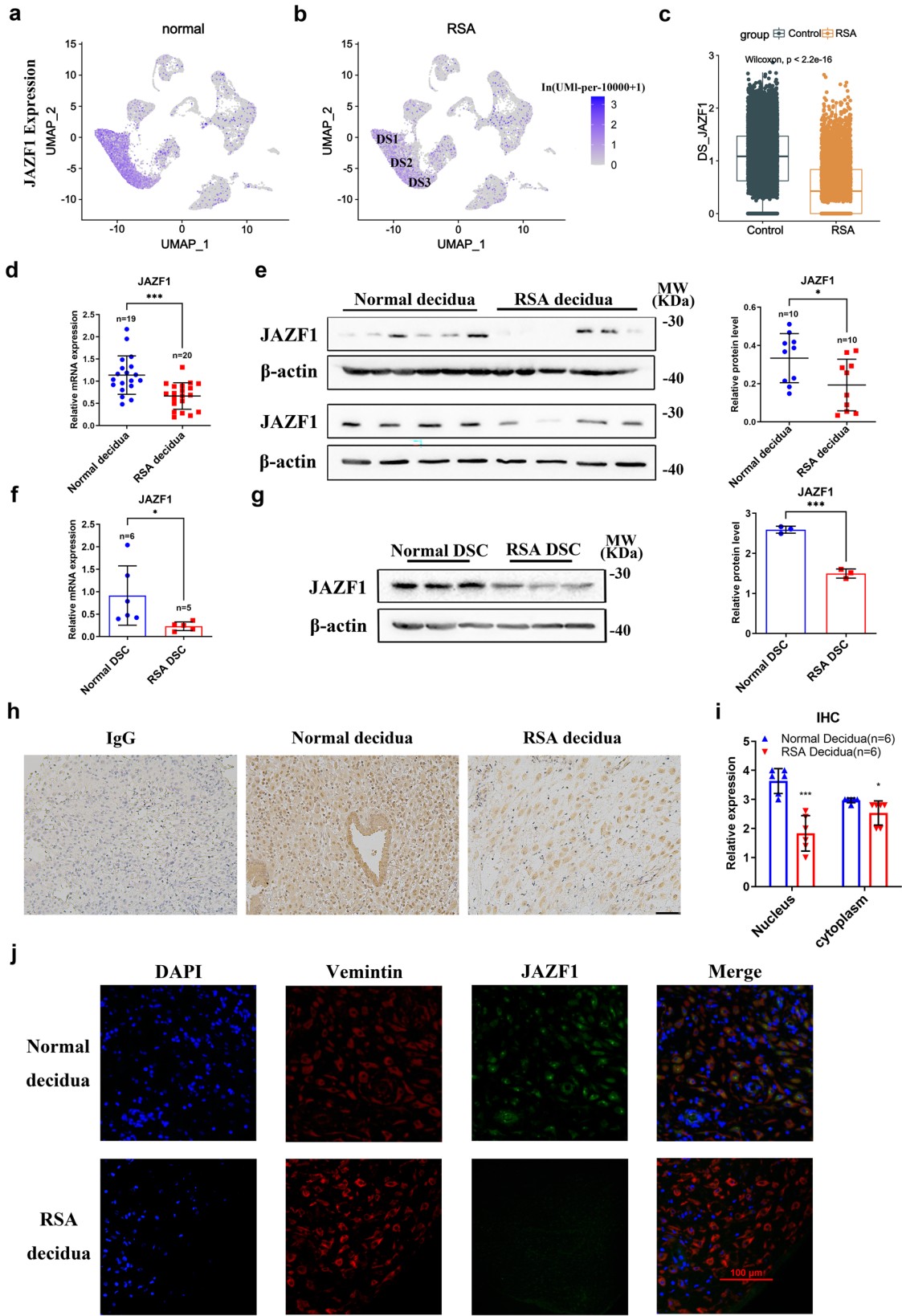

with *JAZF1* knockdown or knockout after decidualization, while BCL2 decreased. (Fig. 3c, d, Supplementary Fig. 2d). Surprisingly, JAZF1 depletion upregulated the expression levels of PRL, IGFBP1, FOXO1, p-FOXO1 and p-FOXO1/FOXO1 (Fig. 3a–d, Supplementary Fig. 2e). At the same time, we observed a noticeable reduction in numbers of decidualized and surviving

cells after JAZF1 knockdown (Supplementary Fig. 2f). Overexpressing *JAZF1* may downregulate the expression of IGFBP1, PRL, BAX/BCL2, BCL2, cyt c, FOXO1, p-FOXO1 and p-FOXO1/FOXO1 in immortalized HESCs decidualized for 6 days, but not BAX, cle-caspase 3 (Fig. 3e–h, Supplementary Fig. 2g–i). Moreover, we observed that the decidualization and

**Fig. 1 Expression of JAZF1 was abnormally attenuated in RSA decidual stromal cells. a, b** UMAP plot showed the mRNA level of *JAZF1* in DS cells was significantly downregulated in RSA decidua (**b**) compared to normal group (**a**). **c** scRNA-seq analysis of *JAZF1* expression in DSCs derived from women with RSA ($n = 6$) and matched normal early pregnancies ($n = 5$) by using ggboxplot function in 'ggpubr' package in R. The data were shown as the median and quartile. **d, e** RT-qPCR (**d**) and Western blotting (**e**) analysis for JAZF1 in normal and RSA decidua. **f, g** RT-qPCR (**f**) and Western blotting (**g**) analysis of JAZF1 in RSA primary DSCs compared to normal group. $n = 3$. **h, i** IHC staining of JAZF1 in the decidua from RSA and normal group, $n = 6$, Scale bar, 100 μm. **j** IF staining of JAZF1 and Vimentin in the decidua from two groups, counterstained with DAPI. $n = 3$, scale bar, 100 μm. Mean ± SD. *$P < 0.05$, ***$P < 0.001$.

apoptosis levels of JAZF1-overexpressed HESCs were downregulated compared with the control group (Supplementary Fig. 2f). The same results were observed in primary ESCs models (Supplementary Fig. 2j, k). We next used flow cytometry to test the anti-apoptotic effect of JAZF1 on decidualized immortalized HESCs. JAZF1 depletion induced cell death via apoptosis while overexpression of *JAZF1* prevented apoptosis (Fig. 3i, j). The cell counting kit 8 (CCK-8) results showed that the proliferative level of decidualized immortalized HESCs was significantly attenuated with *JAZF1* knockdown and upregulated with *JAZF1* overexpression (Fig. 3k). To confirm the relationship between apoptosis and decidualization defects both induced by JAZF1 knockdown, we added a pan-caspase inhibitor (20 μM Z-VAD-FMK) after JAZF1 knockdown and decidualized for 6 days. The results indicated that inhibition of caspase activity can rescue the decidualization defects induced by JAZF1 knockdown (Supplementary Fig. 2l, m). Collectively, these results provide direct evidence that JAZF1 prevents overactivation of the mitochondrial apoptosis pathway and hyper-response of hESCs during the decidualization process.

Some studies suggested that a disordered decidual response or excessive decidual IGFBP1 may be associated with miscarriage, impaired fetal growth or spontaneous preterm delivery[32–34]. Furthermore, premature decidualization during the luteal phase, reflected by PRL expression, may lead to RIF[35]. We further evaluated the impact of decidualized HESCs after *JAZF1* knockdown on the invasion of trophoblast. The transwell assay showed that decidualized HESCs or primary ESCs with *JAZF1* knockdown inhibited HTR-8/SVneo cells invasion and overexpression of *JAZF1* in immortalized HESCs increased trophoblast invasion (Supplementary Fig. 3m), proving the adverse effect of JAZF1 depletion on stromal function during decidualization.

**Apoptosis was upregulated in RSA decidual stromal cells.** In light of the strong pro-apoptotic effect of JAZF1 downregulation observed in decidualization process in vitro, we further assessed the apoptotic level in decidua tissue, as well as the primary DSCs from RSA and control group. Our previous scRNA-seq analysis of DSCs[7] revealed that pro-apoptotic gene *BAX* was obviously increased in DS of RSA group, while *BCL2* did not change significantly (Fig. 4a, Supplementary Fig. 4a). Additionally, GO enrichment analysis of upregulated genes in RSA compared to control group also indicated that cell apoptotic and death pathways were significantly activated in DS1, DS2, and DS3 (Fig. 4b). We also found that obvious upregulation of pro-apoptotic genes BAX, BAX/BCL2, cyt c, cleaved-caspase 3 with downregulation of anti-apoptotic gene BCL2 in RSA decidua in mRNA and protein levels (Fig. 4c–e, Supplementary Fig 4b). Furthermore, the same pattern was observed in primary DSCs isolated from RSA decidua (Fig. 4f, g, Supplementary Fig. 4c). IHC and TUNEL staining further confirmed these observations (Fig. 4h, i). Interestingly, we noticed that the expression of BCL2 positively correlated to

the expression levels of JAZF1, while the opposite trend was observed in cleaved-caspase 3 (Supplementary Fig. 4d). Taken together, apoptosis level was upregulated in DSCs from RSA.

**JAZF1 repressed G0S2 transcription in decidualized hESCs.** To further uncover the molecular mechanisms of induced cell death led by JAZF1 depletion in decidualized hESCs, we performed RNA-seq analysis on immortalized HESCs transfected with si*JAZF1* and OE-*JAZF1*, along with their corresponding controls, after being cultured in differentiation medium for 6 days. We found that pro-apoptotic and decidualization marker genes were significantly upregulated after *JAZF1* knockdown and downregulated with *JAZF1*-overexpression, including *IGFBP1*, *PRL*, *BAX*, *BCL2*, as well as other genes critical for decidualization, such as *FOXO1*, *LEFTY2*, *WNT4*, *PGR*, *WNT5A* (Fig. 5a). GO annotation showed that the differentially expressed genes upregulated by *JAZF1* knockdown were related to the regulation of apoptotic signaling pathway (Fig. 5b). Moreover, we overlapped the upregulated genes with si*JAZF1* and the downregulated genes with *JAZF1*-overexpression (Supplementary Fig. 5a). GO enrichment analysis illustrated these genes were also associated with positive regulation of apoptotic process (Supplementary Fig. 5b). Then we applied CUT & Tag to detect JAZF1-DNA interaction status in undecidualized and decidualized HESCs. Our results showed that there was a total of 21,642 JAZF1-binding peaks in decidualized HESCs with most of them enriched in promoters, distal intergenic and intron (Supplementary Fig. 5c). As shown by peak enrichment and the heatmap of peak distribution, JAZF1-binding was primarily close to TSS in both undecidualized and decidualized HESCs (Supplementary Fig. 5d, e).

Then the overlapped genes between RNA-seq and CUT & Tag were performed GO enrichment analysis and the positive regulation of apoptotic signaling pathway was significantly enriched, including the *G0S2*, *IL19*, *TRIB3* and *BCL2L11* gene, of which *G0S2* was most significantly upregulated after *JAZF1* knockdown (Fig. 5c, d, Supplementary Fig. 5f). Based on the CUT & Tag data obtained from undecidualized and decidualized stromal cells, we found that there were intensive potential JAZF1-binding peaks at the promoter regions of *G0S2* (Fig. 5e).

To further investigate whether G0S2 was activated following *JAZF1* knockdown, we examined the mRNA and protein levels of G0S2 in decidualized HESCs with *JAZF1* knockdown or overexpression. As shown in Fig. 5f, g and Supplementary Fig. 5g, *JAZF1* knockdown increased the mRNA and protein levels of G0S2, but the expression after *JAZF1* overexpression did not change obviously. The upregulation of G0S2 was also observed in primary ESCs models (Supplementary Fig. 4g, h). Consistent with in vitro immortalized HESCs results and increased apoptosis in RSA samples, our previous scRNA-seq analysis of DSCs[7] also revealed elevated *G0S2* in DSCs from RSA group (Fig. 5h). This was further demonstrated by the upregulated mRNA and protein levels of G0S2 in RSA decidua

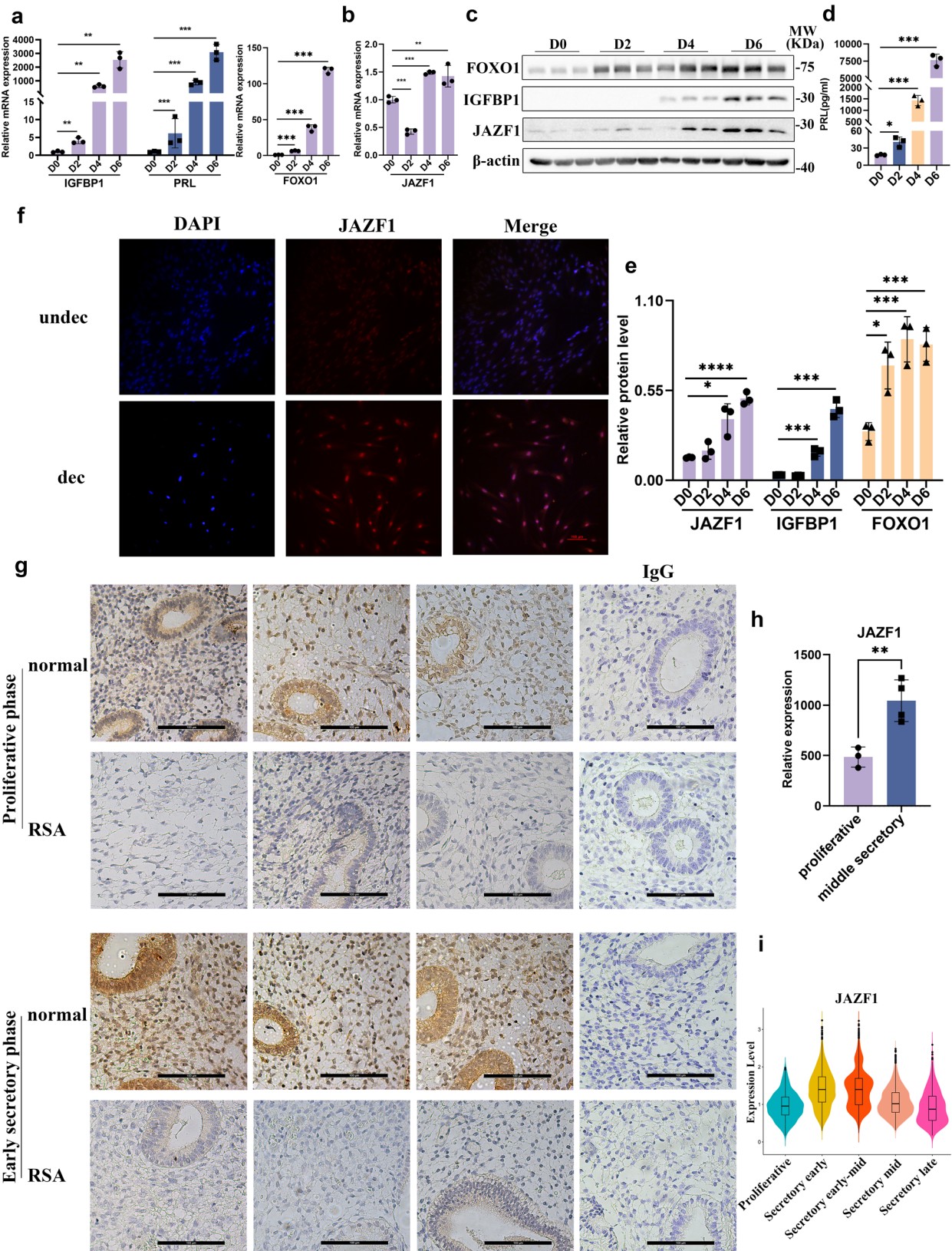

and primary DSCs compared to normal group (Fig. 5i–l), as well as by the IHC results (Fig. 5m). We noticed that the expression of G0S2 negatively correlated to the expression levels of JAZF1 (Supplementary Fig. 4i). In summary, JAZF1 inhibited G0S2 expression in decidualized hESCs, while G0S2 was upregulated in decidua of RSA women.

**JAZF1 knockdown induced apoptosis and decidualization defect of hESCs by activating G0S2.** Several studies reported that overexpression of *G0S2* promoted cell death, such as in M1 macrophages[36]. To verify whether G0S2 upregulation accounts for the main cause of apoptosis in JAZF1 deficient stromal cells, we overexpressed *G0S2* by lentivirus in immortalized HESCs as

**Fig. 2 JAZF1 is dynamically expressed in human endometrial stromal cells. a–e** RT-qPCR (**a**, **b**), Western blotting (**c**, **e**), and ELISA (**d**) analysis for JAZF1, IGFBP1, PRL, FOXO1 in immortalized HESCs treating with MPA, cAMP for 6 days. **f** IF staining of decidualized immortalized HESCs for JAZF1, counterstained with DAPI. undec: undecidualized HESCs; dec: decidualized HESCs. $n = 3$, scale bars, 100 μm. **g** IHC staining of JAZF1 in the proliferative and early secretory phase endometrium from RSA and normal group. $n = 3$, scale bars, 100 μm. **h** Expression of JAZF1 in proliferative and middle secretory phase endometrium. The data were retrieved from microarray data deposited in the Gene Expression Omnibus (GDS2052). **i** Vlnplot of the expression level of *JAZF1* in ESCs across the natural menstrual cycle identified by single-cell transcriptomics. The data were retrieved from microarray data deposited in the Gene Expression Omnibus (GSE111976). The data were shown as the median and quartile. Others were shown as mean ± standard deviation (SD). *$P < 0.05$, **$P < 0.01$, ***$P < 0.001$.

efficiency was verified by qPCR and western blotting (Fig. 6a, b). Overexpression of *G0S2* significantly increased the expression of BAX, BAX/BCL2, cleaved-caspase 3, cyt c and FOXO1 and decreased BCL2 and p-FOXO1/FOXO1, but not p-FOXO1 (Fig. 6c, d, Supplementary Fig. 6a, b). In addition, the mRNA and protein levels of IGFBP1 were significantly upregulated and the mRNA level of *PRL* was slightly upregulated after over-expression of *G0S2* (Fig. 6c, d). Furthermore, flow cytometry also indicated *G0S2* overexpression triggered significant cell death in decidualized HESCs (Fig. 6e). As expected, proliferation of decidualized HESCs was significantly downregulated with *G0S2* overexpression (Fig. 6f). Similarly, 20 μM Z-VAD-FKM was added to cells overexpressing G0S2 and decidualized to detect decidualization levels. The results also suggested that caspase inhibitors may alleviate decidualization defects caused by over-expression of G0S2 (Supplementary Fig. 6c–e). These results indicated G0S2 was a crucial regulator of cell death via mitochondrial apoptosis and decidualization of immortalized HESCs.

To explore whether the downregulation of G0S2 restored the apoptotic defects of immortalized HESCs or primary ESCs caused by JAZF1 depletion, we inhibited *G0S2* using siRNA and chemical inhibitor (NS-3) (Supplementary Fig. 7a–e for siRNA and inhibitor silencing efficiency). Treatment with the G0S2 inhibitor NS-3, combined with decidualization induction for 6 days, efficiently attenuated the expression of BAX, BAX/BCL2, cleaved-caspase 3, cyt c and upregulated BCL2 in JAZF1-deficient cells, but not FOXO1, p-FOXO1 and p-FOXO1/FOXO1 (Fig. 7a, b, Supplementary Fig. 7f–h). Furthermore, treated with NS-3 obviously decreased the upregulation of IGFBP1, but not PRL (Fig. 7a–c), suggesting that JAZF1 depletion promoted cell death via mitochondrial pathway and IGFBP1 expression deficiency of immortalized HESCs during the decidualization process by activating G0S2.

In addition, we validated these findings by using siRNA to simultaneously knock down *JAZF1* and *G0S2* (Fig. 7d–f, Supplementary Fig. 7f–h). The same expression patterns were observed in primary ESCs models (Supplementary Fig. 7i–k). Flow cytometry also demonstrated that knockdown of *JAZF1* and inhibition of G0S2 effectively rescued the increased apoptosis of JAZF1-deficient cells (Fig. 7g, h). Additionally, attenuated proliferation of decidualized HESCs after *JAZF1* knockdown was rescued by inhibition of G0S2 (Fig. 7i). Collectively, our in vitro evidence strongly suggested JAZF1 depletion triggered cell death via mitochondrial apoptosis pathway and excessive upregulation of IGFBP1 of decidualized hESCs by activating G0S2.

**The JAZF1 repressed G0S2 transcription by restricting the activity of its upstream activator Purβ.** To further determine how JAZF1 regulates the transcriptional level of G0S2, we performed immunoprecipitation (IP), mass spectrometry (MS) analysis and a high-throughput protein array screening on JAZF1 interacting partners. We screened out the transcription factor Purβ from the results of MS analysis and protein array (Fig. 8a, Supplementary Data 2). We confirmed the interaction between

JAZF1 and Purβ in decidualized immortalized HESCs by co-IP (Fig. 8b). To further confirm the function of Purβ in immortalized HESCs and whether it regulates G0S2, we performed *Purβ* knockdown in immortalized HESCs using siRNAs targeting *Purβ* (Supplementary Fig. 8a, d for siRNA silencing efficiency). The knockdown of *Purβ* downregulated the expression of G0S2, BAX, BAX/BCL2, cleaved-caspase 3, cyt c and rescued the anti-apoptotic gene BCL2 (Fig. 8c, d, Supplementary Fig. 8b) in si*JAZF1* immortalized HESCs after decidualization. Consistently, the expression of IGFBP1 and PRL was significantly rescued with *Purβ* knockdown during si*JAZF1* immortalized HESCs decid-ualization, but not FOXO1, p-FOXO1 and p-FOXO1/FOXO1 (Fig. 8c, d, Supplementary Fig. 8c). Similar expression patterns were observed in two primary ESCs models (Supplementary Fig. 8d–g). Furthermore, flow cytometry demonstrated that knockdown of *JAZF1* and inhibition of *Purβ* obviously rescued the upregulated level of apoptosis in JAZF1-deficient cells (Fig. 8e). The above results tentatively suggested that *Purβ* knockdown could inhibit G0S2 transcription as well as mito-chondrial apoptosis pathway in decidualized hESCs. Next, we detected the localization of Purβ in decidua by IHC. The results indicated that Purβ localized in the nucleus and cytoplasm of DSCs, and its expression of RSA was higher than that of the control group (Fig. 8f), which further suggested that Purβ may regulate G0S2.

We then further confirmed whether Purβ controlled the *G0S2* promoter activity by using a dual luciferase reporter assay and the results showed that Purβ increased *G0S2* promoter-driven luciferase activity in a dose-dependent manner (Fig. 8g). Importantly, Purβ significantly upregulated *G0S2* promoter-driven luciferase activity with *JAZF1* knockdown, which demon-strated that JAZF1 inhibited the function of Purβ on the transcriptional activation of *G0S2* (Fig. 8h). In summary, the JAZF1 maintains decidua homeostasis by restricting the tran-scription of *G0S2* activated by Purβ, thereby preventing cell death via apoptosis as well as decidualization defects.

**Discussion**
RSA is a common complication of pregnancy with multiple etiologies[10,37], including chromosomal errors, immune system abnormalities and decidualization defects of endometrial stromal cells During decidualization, cells undergo mitosis, differentiation and apoptosis, and the balance of these processes is indispensable for successful implantation in both humans[38] and animal models[10]. Notably, accumulating studies have indicated that excessive cell apoptosis is one of the leading causes of RSA[15,17]. In the present study, we demonstrated that JAZF1 represses G0S2 expression by restricting its transcription activator Purβ, thereby limiting the activation of themitochondrial apoptotic pathway and controlling the expression of decidualization markers such as IGFBP1, thus governing the survival and function of hESCs (Fig. 9). The pattern of impaired JAZF1, high G0S2, excessive apoptosis and decid-ualization defects were consistently observed in RSA decidua.

During decidualization, apoptosis is accompanied by pro-liferation and differentiation of the ESCs[38,39] which should be

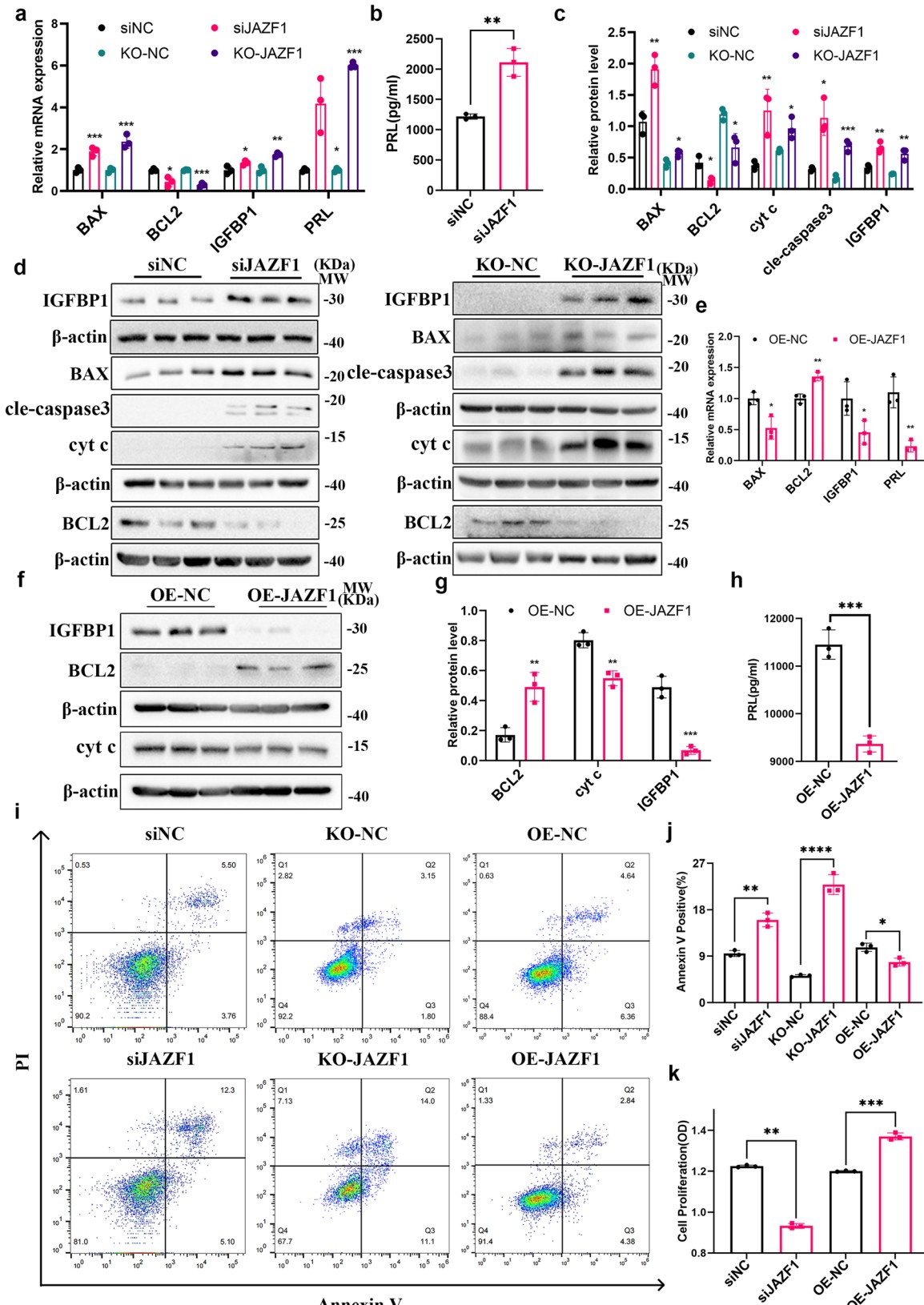

tightly controlled. Excessive apoptosis of ESCs during decidualization can cause decidua defects, leading to abortion and even RSA[40]. It is reported that upregulation of CDKN1A, BAX, CCL28 elevate apoptosis in decidualized hESCs and are strongly associated with RSA[16,41]. In addition, anandamide induces a decrease in rat decidual cells viability followed by mitochondrial stress and ROS production, leading to apoptosis and ultimately decidua defects[42]. Here, we further proved the relationship between the cell death via excessive apoptosis and decidua defects as observed in RSA endometrium. More importantly, we identified JAZF1 as a transcriptional regulator that prevents excessive apoptosis of hESCs and governs the function of the decidua during pregnancy.

**Fig. 3 JAZF1 depletion promoted apoptosis and decidualization defect of stromal cells. a** Results of RT-qPCR showing the expression of *IGFBP1*, *PRL*, *BAX*, *BCL2* in HESCs after *JAZF1* knockdown and knockout. $n = 3$. **b** ELISA detection showed the PRL level after *JAZF1* knockdown. $n = 3$. **c, d** Protein expression of IGFBP1, BAX, BCL2, cleaved-caspase 3, cyt c in HESCs with transfection of si*JAZF1* or KO-*JAZF1* during decidualization. $n = 3$. **e** RT-qPCR analysis for *IGFBP1*, *PRL*, *BAX*, *BCL2* in JAZF1-overexpression HESCs followed by culture in medium supplemented with MPA and cAMP for 6 days. $n = 3$. **f, g** Western blotting analysis of IGFBP1, JAZF1, BCL2, cyt c in decidualized HESCs transfected with OE-NC or OE-*JAZF1* lentivirus. $n = 3$. **h** ELISA detection showed the PRL level after *JAZF1* overexpression. $n = 3$. **i, j** Flow cytometry of decidualized HESCs transfected si*JAZF1* and overexpression-*JAZF1* lentivirus. $n = 3$. **k** Cell proliferation was determined by CCK8 assay. $n = 3$. The data were shown as the mean ± standard deviation (SD). *$P < 0.05$, **$P < 0.01$, ***$P < 0.001$.

To be noted, besides the cell death, we found that the decidualization markers IGFBP1 and PRL were upregulated after *JAZF1* knockout in decidualized HESCs which also were observed in RSA decidua[43]. These excessively decidualized HESCs impaired the invasion of HTR-8/SVneo cells, suggesting the functions of these immortalized HESCs are still defective. In consistent with this observation, it is reported that IGFBP1 inhibited cytotrophoblast invasion into differentiated endometrial stroma in vitro[44]. Experiments with IGFBP1 transgenic mice also revealed that excessive decidual IGFBP1 was linked to abnormal placental development and impaired fetal growth[32]. Furthermore, several data from the placenta and cervical fluid suggested that elevated decidual IGFBP1 level triggered spontaneous preterm delivery and intrauterine growth restriction[34,45,46]. More importantly, premature advancement of decidualization during the luteal phase, reflected by PRL expression, may lead to embryo-endometrial asynchrony resulting in RIF[47]. These findings suggested that excessive IGFBP1 and premature decidualization might exacerbate negative consequences during pregnancy. It's recommended that more comprehensive studies should be taken to determine the decidua defects besides the IGFPB1 and PRL expression.

Besides the endometrial stromal cells, the anti-apoptosis role of JAZF1 has been found in many other cell types. Knockdown of *JAZF1* in pancreatic β-cells resulted in significant activation of the endoplasmic reticulum apoptotic pathway and obvious upregulation of apoptosis[25]. In addition, silencing of *JAZF1* led to a downregulation of cell proliferation and DNA damage as well as an enhancement of the early and late apoptosis of human spermatogonial stem cells[26]. Different from other cells, we found knockdown of *JAZF1* led to immortalized HESCs or primary ESCs cell death through mitochondrial apoptosis pathway indicated by the upregulation of BAX and down-regulation of BCL2, suggesting distinct pathways were involved in the anti-apoptosis function of JAZF1 depending on the cell types.

JAZF1 is a multifunctional regulatory factor that was initially discovered as TR4 corepressor and it inhibits the physiological function of TR4 by interacting specifically with the ligand-binding structural domain of *TR4*[21]. Additionally, JAZF1 represses SREBP-1c expression by inhibiting the transcriptional activity of the liver X receptor response element (*LXRE*) in the *SREBP-1c* promoter[22]. These findings indicate that JAZF1 is a transcriptional repressor. Here, in HESCs we found JAZF1 interacts with transcription factor Purβ and repress its activity on G0S2 transcription to prevent cell death in immortalized HESCs during decidualization. JAZF1 binds to the *G0S2* gene locus and knockdown of *JAZF1* increased Purβ activation of G0S2 as shown by luciferase assay. Our results proved that JAZF1 worked as a typical transcription repressor just like in other systems. However, JAZF1 targeted apoptosis-related gene G0S2 in immortalized HESCs during decidualization, implying cell type specific functions of JAZF1. Moreover, as a transcription repressor, whether JAZF1 directly regulates the transcription of key decidualization-related genes requires further experiments.

Previous studies have found that G0S2 induces cell apoptosis by binding to BCL2 and activating mitochondrial apoptosis pathway[36,48]. In the current study, we found that G0S2 was a molecule that regulated stromal cells survival and decidualization. Mechanistic studies led to the discovery that JAZF1 depletion induced cell death via mitochondrial apoptosis in stromal cells through activation of G0S2. However, when *JAZF1* and *G0S2* were co-downregulated, they only down-regulated IGFBP1 expression, but not PRL expression, which suggests that PRL may be regulated by other downstream networks of JAZF1. Transcription factor Purβ has been found to bind to DNA promoter sequences of several genes[49,50] and thus participate in transcriptional regulation, but the effect of its expression on cell function has not been described. This study showed that Purβ could bind to *G0S2* promoter and promote its transcription, thus inducing cell death and decidualization defects. More importantly, knockdown of *Purβ* or *G0S2* restored the apoptotic defects of *JAZF1*-knockdown HESCs or primary ESCs.

In spite of the strengths, there were some limitations that should be paid attention to. Firstly, the sample size in this paper was limited due to the complexity and heterogeneity of RSA etiology. Although we observed consistent results in hESC, primary ESC and decidua tissues possibly due to the strict rules we chose samples, a larger sample scale will further nail down the conclusions we got. In addition, decidualization involves multiple cell types in decidual tissue, so the contribution of other cells to decidualization cannot be excluded. In this study, we focused mainly on stromal cells alone. Further studies are needed to explore the role of cell interaction in decidualization. Moreover, the results of this study were only verified at the cellular level, the use of conditional gene knockout animal models in the future will help validate our observation in cultred ESCs.

Collectively, our investigation provides compelling evidence that JAZF1 plays a key role in hESCs survival and decidua homeostasis through the transcriptional regulation of critical factors related to cell death and decidualization. JAZF1 inhibits the transcription of G0S2 by binding to its promoter and restricting the activity of transcription factor Purβ, thereby preventing hESCs from cell death via apoptosis and maintaining successful pregnancy. A better understanding of the regulatory network of JAZF1 will facilitate the development of therapeutic strategy for the clinical treatment of RSA.

## Methods

**Sample of clinical cases**. Healthy women with elective abortions without medical indication and RSA women with two or more consecutive spontaneous unexplained abortions were recruited from the Third Affiliated Hospital of Guangzhou Medical University in China from January 2020 to May 2022 (gestational age 6–9 weeks). The age of participates is between 20 and 35 years old. Patients with endocrine disorders, metabolic, or autoimmune diseases were excluded. Only patients with unexplained RSA were recruited for the study. The study had been approved by the Medical Ethics Committee of The Third Affiliated Hospital of Guangzhou Medical University, Medical Research (No.2020127) and all participants signed informed consent. Detailed information of patients is listed in Supplementary Table 1–4.

**Isolation and culture of primary decidual stromal cells**. Primary DSCs were obtained from normal pregnancy and RSA. The isolation and culturing of primary

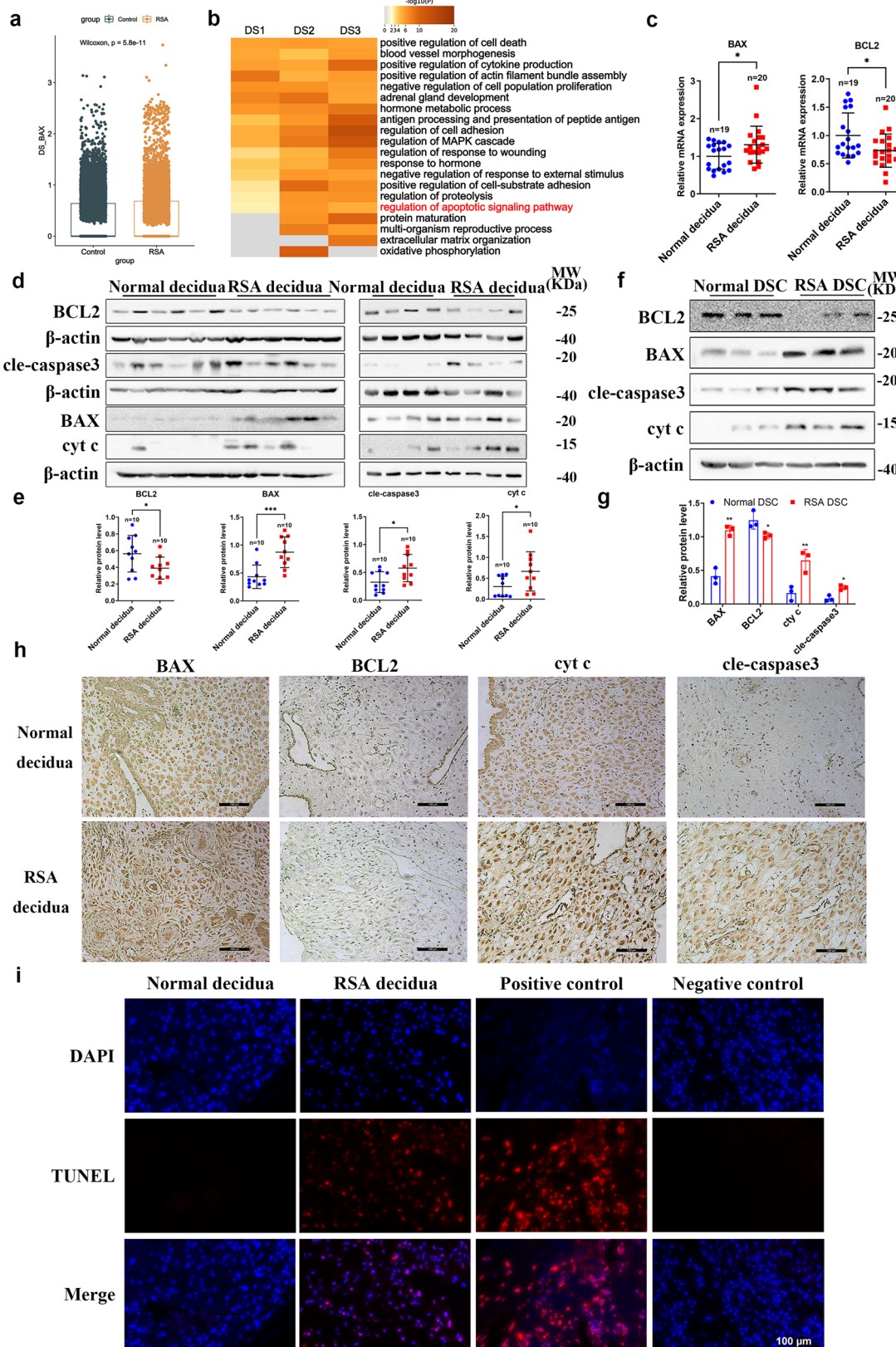

**Fig. 4 Apoptotic level was upregulated in RSA decidual stromal cells. a** scRNA-seq analysis of *BAX* expression in DSCs. The data were shown as the median and quartile. **b** GO enrichment analysis of DS1, DS2, DS3. **c–e** RT-qPCR (**c**) and Western blotting (**d**, **e**) analysis of BAX, BCL2, cleaved-caspase 3, cyt c in normal and RSA decidua. **f**, **g** Western blotting analysis of BAX, BCL2, cleaved-caspase 3, cyt c in RSA DSCs compared to normal group. $n = 3$. **h** IHC analysis of BAX, BCL2, cleaved-caspase 3, cyt c in RSA. $n = 3$, scale bars, 100 μm. **i** TUNEL staining was used to measure the decidual apoptosis in each group. $n = 4$, scale bars, 100 μm. The data were shown as the mean ± standard deviation (SD). *$P < 0.05$, **$P < 0.01$, ***$P < 0.001$.

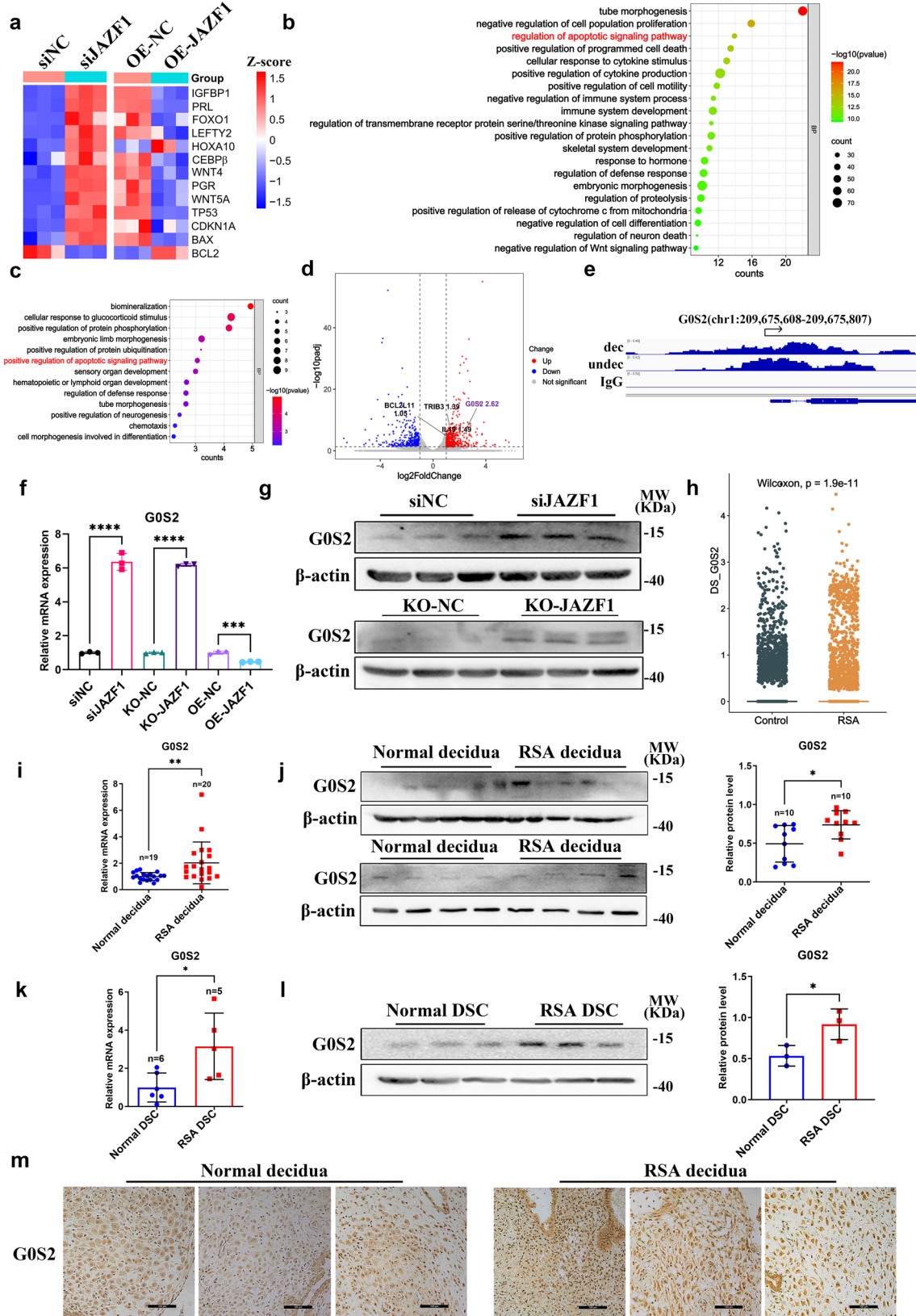

decidual stromal cells were performed as follows. The decidual tissues were first cut into pieces as small as possible and subjected to 1 mg/ml type IV collagenase digestion at 37 °C for 1.5 h. After adding 2% medium (DMEM/F12, 1%P/S, 2% charcoal-stripped fetal bovine serum (CS-FBS, Biological Industries)) to terminate digestion, the cell suspension was filtered through 100 μm and 40 μm cell strainers, respectively, and then was centrifuged at 1000 rpm for 5 min. After using red blood cell lysis, cells were seeded in a 10 cm dish with $10^7$ cells and cultured in DMEM/

F12 complete medium supplemented with 1% P/S and 10% CS-FBS overnight. DSCs were collected the next day to extract mRNA and protein for subsequent experiments.

**Isolation and culture of primary endometrial stromal cells.** Primary ESCs were obtained from two female patients with tubal infertility and no endometrial

**Fig. 5 JAZF1 directly targeted to G0S2 in decidualized hESCs, of which expression was increased in decidua in RSA. a** Heatmap of apoptosis-related and decidualization-related genes from RNA-seq after *JAZF1* knockdown and overexpression in decidualized HESCs. *n* = 3. **b** GO analysis of the differentially expressed genes (DEGs) in RNA-Seq. **c** GO analysis of overlapped genes of JAZF1 binding and *JAZF1* upregulated genes. **d** DEGs including *G0S2, IL19, TRIB3* and *BCL2L11* detected by RNA-seq of immortalized HESCs after decidualization induction for 6 days in control and *JAZF1* knockdown samples as visualized by volcano plot. **e** Visualization of JAZF1 binding peaks on *G0S2* locus. Reference genome:hg38; Undec: undecidualized HESCs; Dec: decidualized HESCs. **f, g** RT-qPCR (**f**) and Western blotting (**g**) detection of G0S2 in si*JAZF1* and KO-*JAZF1* decidualized HESCs. *n* = 3. **h** scRNA-seq analysis of *G0S2* expression in DSCs derived from women with RSA (*n* = 6) and matched normal early pregnancies (*n* = 5) by using ggboxplot function. The data were shown as the median and quartile. **i, j** mRNA (**i**) and protein (**j**) levels of G0S2 in RSA decidua compared to normal pregnancy. **k, l** mRNA (**k**) and protein (**l**) levels of G0S2 in primary DSCs from normal pregnancy and RSA. *n* = 3. **m** IHC analysis of G0S2 in RSA decidua compared to normal decidua. *n* = 4. scale bars, 100 μm. The data were shown as the mean ± standard deviation (SD). **P* < 0.05, ***P* < 0.01, ****P* < 0.001, *****P* < 0.0001.

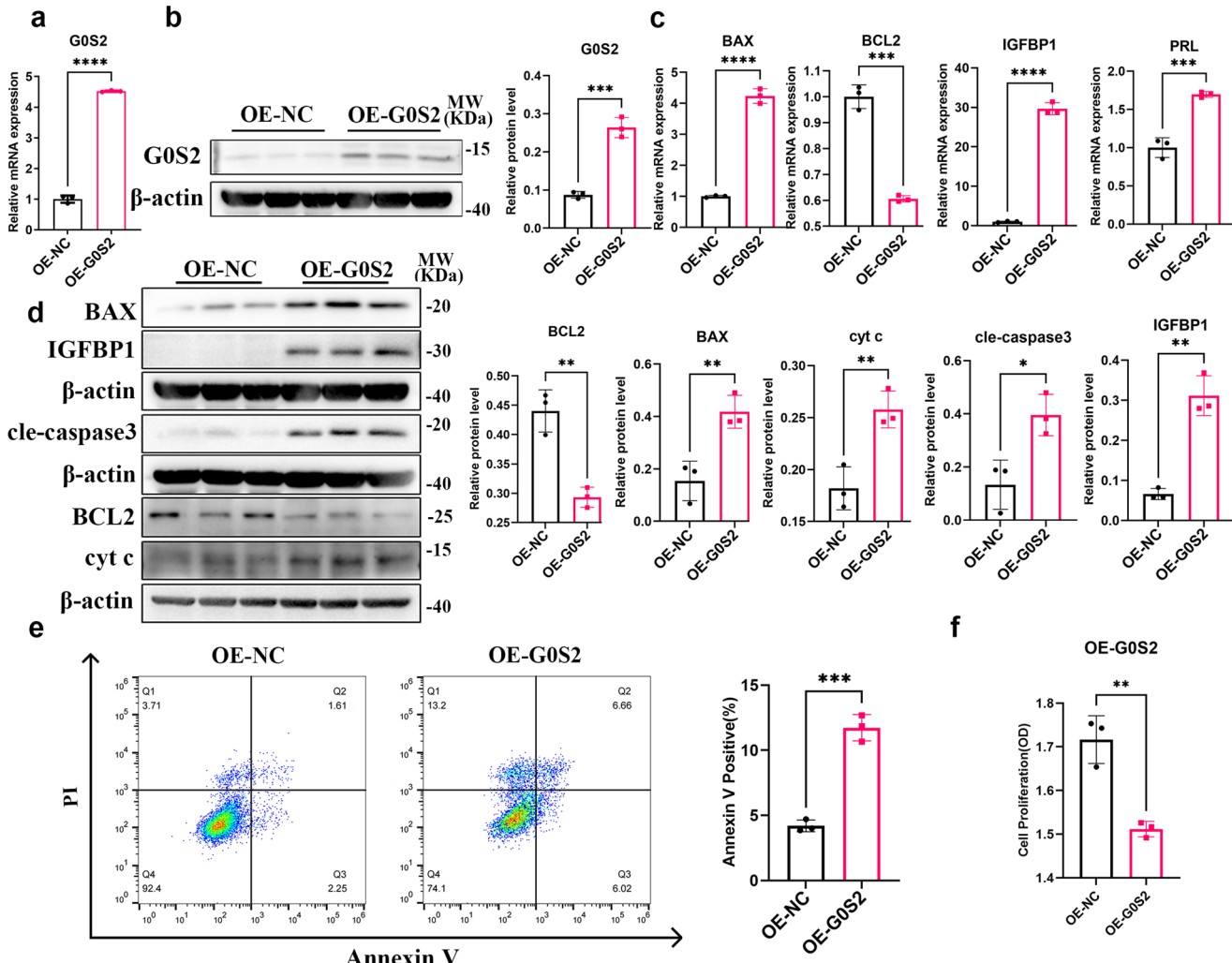

**Fig. 6 Overexpression of G0S2 activated mitochondrial apoptosis pathway and decidualization defect in decidualized hESCs. a, b** G0S2 mRNA and protein levels in HESCs with infected with control lentivirus or *G0S2*-overexpression lentivirus. **c** RT-qPCR analysis for *IGFBP1, PRL, BAX, BCL2* in G0S2-overexpression HESCs followed by cultured in medium supplemented with MPA and cAMP for 6 days. *n* = 3. **d** Western blotting analysis of IGFBP1, BAX, BCL2, cleaved-caspase3, cyt c in decidualized HESCs transfected with OE-NC or OE-*G0S2* lentivirus. *n* = 3. **e** Flow cytometry of decidualized HESCs transfected with control lentivirus or overexpression-*G0S2* lentivirus. *n* = 3. **f** Cell proliferation was determined by CCK8 assay. *n* = 3. The data were shown as the mean ± standard deviation (SD). **P* < 0.05, ***P* < 0.01, ****P* < 0.001, *****P* < 0.0001.

pathology abnormalities. Detailed information of patients was listed in Supplementary table 5. The isolation and culturing of primary ESCs were performed as previously described[51]. (1) The tissue after cutting was placed in 20 mL of enzymatic digestion solution (DMEM/F12 supplemented with 1% P/S, 0.4 mg/mL Collagenase V, 1.25 U/mL dispase II) and incubated at 37 °C for 40–50 min. (2) After adding the same amount of complete medium, the digested tissues were centrifuged at 350 rpm for 5 min. (3) The supernatant was filtered through a 100 μm filter and centrifuged again. The cells were resuspended in complete medium and planted in a 10 cm dish. Primary ESCs were maintained in DMEM/F12 without phenolic red (Biological Industries)

in the presence of 10% CS-FBS (Biological Industries), 1% penicillin-streptomycin (Gibco).

**Establishment of JAZF1 knockout cell lines.** Immortalized human endometrial stromal cell line was a gift from Pro. Haibin Wang (Xiamen University, ATCC CRL-4003). *JAZF1* knockout was generated by CRISPR/Cas9 approach as previously described[52]. (1) sgRNAs targeting *JAZF1* gene (NM_175061.4) were designed on the website (https://portals.broadinstitute.org/gpp/public/) and cloned into the pLentiCRISPRv2 Neo vector (Addgene plasmid #127644). (2)

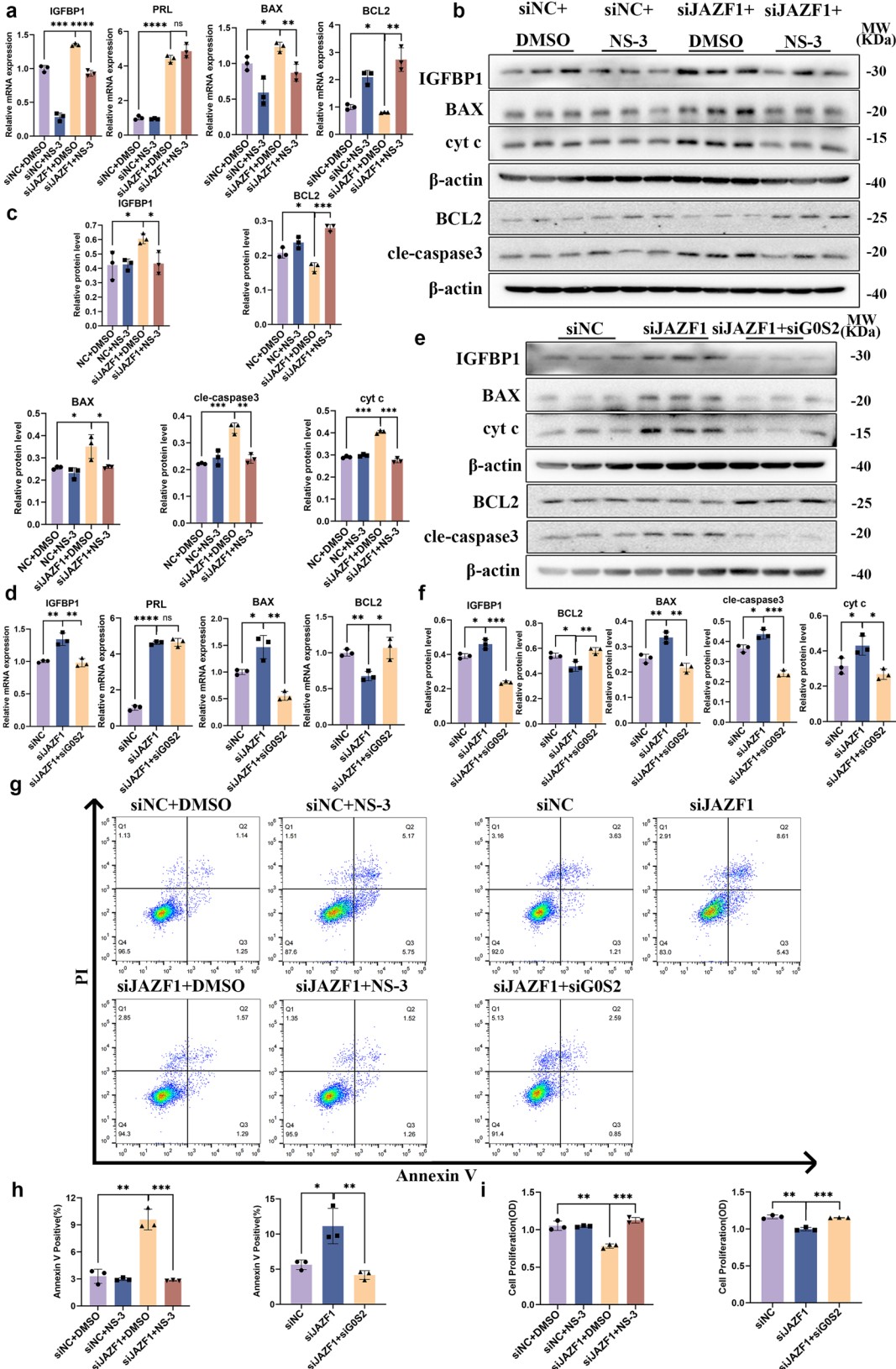

**Fig. 7 JAZF1 depletion promoted cell death of decidualized HESCs by activating G0S2. a** Results of RT-qPCR showing the expression of *IGFBP1*, *PRL*, *BAX*, *BCL2* in immortalized HESCs transiently transfected with si*JAZF1* and treated with G0S2 inhibitor NS-3. *n* = 3. **b, c** Protein expression of IGFBP1, BCL2, BAX, cleaved-caspase 3, cyt c, G0S2 in immortalized HESCs with transfection of si*JAZF1* and added G0S2 inhibitor NS-3 during decidualization. *n* = 3. **d** RT-qPCR analysis for *IGFBP1*, *PRL*, *BAX*, *BCL2* in siNC, si*JAZF1* and si*JAZF1*+si*G0S2* immortalized HESCs after decidualization. *n* = 3. **e, f** Western blotting analysis of IGFBP1, BAX, cleaved-caspase 3, cyt c, G0S2 in decidualized HESCs transfected with si*JAZF1*+si*G0S2*. *n* = 3. **g, h** Flow cytometry of decidualized HESCs transfected si*JAZF1* + NS-3 or si*JAZF1*+si*G0S2*. *n* = 3. **i** Cell proliferation was determined by CCK8 assay. *n* = 3. The data were shown as the mean ± standard deviation (SD). *P < 0.05, **P < 0.01, ***P < 0.001.

CRISPR/Cas9 plasmid was packaged with lentivirus. After infecting HESCs, neomycin positive cells were sorted and plated into a 96-well plate with one cell per well. (3) Knockout efficiency was verified by DNA sequencing, RT-qPCR and western blotting after single-cell-derived clone expansion. Detailed information of sgRNA and plasmid are listed in supplementary table 6 and 7, respectively.

**In vitro culture of HESCs and decidualization.** The culture system of immortalized HESCs was consistent with that of primary ESCs. For decidualization, immortalized HESCs were cultured in medium at the present of medroxyprogesterone acetate (MPA, 4 µM, Aladdin), and dibutyl cyclophospsinoside (db-cAMP, 1 mM, Sigma) in 2% CS-FBS with different days. All the cells were cultured in 5% $CO_2$ at 37 °C and culture medium was changed every 2 days. Details of the reagents used are in the supplementary Data 1.

**Plasmid construction and siRNA transfection.** We performed PCR primer design by combining the *JAZF1* or *G0S2* CDS region queried on NCBI and the multiple cloning sites of the overexpression plasmid. We next performed PCR amplification of *JAZF1* or *G0S2* according to the instructions of GXL (Takara), and the amplified products were purified by agarose gel electrophoresis and cloned into the pLVX-IRES-Neo vector. The list for plasmid used in this study is shown in Supplementary table 6. The lentivirus was packed with HEK 293 T cells (ATCC, USA). 72 h after transfection, the virus was filtered by 0.45 µm filter and concentrated with Lenti-X Concentrator (Takara) which was used to infect immortalized HESCs respectively for 48 h. After transfection, we selected the cells with neomycin at a concentration of 1.3 mg/ml. siRNAs targeting *JAZF1*, *G0S2* and *Purβ* were purchased from Suzhou GenePharma Company (see the specific sequence for details in Supplementary table 6). RNA interference was carried out according to the manufacturer's instructions. Briefly, 20 µM siRNA was transfected into immortalized HESCs with Lipofectamine RNAi MAX (Invitrogen, Carlsbad, USA). Both transfection of immortalized HESCs underwent RT-qPCR and western blotting to confirm the knockdown and overexpression efficiency. For the knockdown and overexpression experiment, the silencing and overexpression were first performed without decidualization stimulus, and 24 h later immortalized HESCs were decidualization for 6 days. Details of the reagents and plasmids used are in the supplementary Data 1 and table 7.

**CUT & Tag.** CUT & Tag experiment was performed according to the manual of CUT & Tag Assay Kit (Vazyme, Cat#TD903). Immortalized HESCs ($1 \times 10^6$) were used for the CUT & Tag experiment. Briefly, after activating conA beads, 95 µ wash buffer was added into samples and incubated for 10 min at room temperature Subsequently, JAZF1 antibody-beads complex incubation was performed on a rotator for 2 h at RT. Followed by secondary antibody incubation, samples were mixed with 1:120 dilution of pA/G-Tnp5 transposase (0.033 mM) for an hour. Next, sample was labeled in 40 ml tagmentation buffer at 37 °C for 1 h. According to the manufacturer's guide, DNA was extracted and dissolved in 20 µ nuclease-free water. To amplify libraries, we added P5/P7 adapter (Vazyme, Cat#TD202), 2× CAM and performed PCR. PCR clean-up was performed by adding 2 volume of DNA clean beads and libraries were eluted in 30 µl nuclease-free water. Samples were pooled and sequenced using 150 bp paired-ended on Nova seq platform. Details of the reagents used are in the supplementary Data 1.

**Luciferase reporter assays.** Human *G0S2* promoter (-2001 to -1) luciferase reporter plasmids and Renilla reporter plasmids were transiently co-transfected with *Purβ* expression plasmid into HEK 293 T cells using jetPRIME reagent. Cells were collected 24 h after transfection, and luciferase activity was measured using a luciferase assay kit (Beyotime, P2179S). Luciferase activity was normalized to Renilla levels. Detailed information of PCR primers is listed in Supplementary table 6. Details of the reagents and plasmids used are in the supplementary Data 1 and table 7.

**Human proteome microarray assay.** The HuProt™ microarray (CDI Laboratories, Inc.) composed of ~20000 human full-length proteins with N-terminal glutathione S-transferase (GST) tags was used to identify JAZF1 interactors. The HuProt™ microarray assay was performed by GeneCloud Biotechnologies Inc. (Guangzhou, China) according to the following procedure. Human Proteome microarrays (HuProt™ 20 K) were blocked with blocking buffer (3 ml 10% BSA in 7 ml 1×PBS) for 3 h at 4 °C. JAZF1 protein (GeneCloud Biotechnologies Inc., Guangzhou, China) was labeled with biotin, then added 3 ml of diluted labeled protein (3 µg protein in 3 ml blocking solution) to each well that contains a blocked HuProt™ microarray and incubated at RT for 1 hr. The microarrays were washed for 10 min with 1×TBST and rinsed for three times with 0.1× TBS. To remove fluid, span the microscope slide box or the 50 ml tubes containing the microarrays in a centrifuge at 800 rpm for 3 min. The microarrays were incubated with streptavidin-Cy5 at 1:1,000 dilution for 1 h at room temperature and underwent three more 5 min washes. It was spun dry at 1500 rpm for 3 min and subjected to scanning with a LuxscanTM 10K-A (Capitalbiotech, Beijing, China) in order for results to be visualized and recorded.

**Immunoprecipitation.** For protein interaction, IP experiments were performed according to the manual of Immunoprecipitation Kit with Protein A + G Magnetic Beads (Beyotime, P2179S). Anti-JAZF1 (Santa Cruz, sc-376503X) and normal mouse IgG were used for IP. The immunoprecipitants were washed three times in lysis buffer in the presence of phosphatase inhibitor and Protease inhibitor cocktail, resolved with sodium dodecyl sulfate–polyacrylamide gel electrophoresis (SDS-PAGE), and immunoblotted with corresponding antibodies. Details of the reagents used are in the supplementary Data 1.

**IP MS.** For protein interaction, IP experiments were performed according to the manual of Immunoprecipitation Kit with Protein A + G Magnetic Beads (Beyotime, P2179S). Cell line immortalized HESCs were used to prepare IP samples after treated with MPA and cAMP for 6 days. After co-IP with JAZF1-conjugated protein A + G magnetic beads, the beads-Ab-Ag complexs were resolved with SDS-PAGE. The discrete bands between JAZF1 and IgG control were isolated, digested, purified, and subjected to liquid chromatograph (LC)-MS in Wininnovate Company. Please refer to Supplementary Data 2 for detailed mass spectrometry results and supplementary Data 2 for antibody concentrations and providers.

**TUNEL assay.** TUNEL assay was performed according to the manual of TUNEL BrightRed Apoptosis Detection Kit (Vazyme, A113). After removal of paraffin with xylene and reconstituting in ethanol solutions of decreasing concentrations, sections were digested for 20 min with proteinase K (20 µg/ml) at room temperature, washed in distilled water. We treated samples with DNase I to prepare positive controls. Subsequently, sections were labled by BrightRed Labeling Mix. Finally, cells were defined as apoptotic if they were red-positive with fluorescence microscope. This experiment was repeated on several different sections for each specimen, obtaining similar results. Details of the reagents used are in the supplementary Data 1.

**Flow cytometry.** Apoptosis of immortalized HESCs transfected with siRNA or overexpression virus were analyzed by Annexin V Apoptosis Detection kit according to the manufacturer's recommendations (Vazyme, A211). FlowJo software (v10.8.1) was used for data analysis and graphic representation. We used FSC and SSC jointly for gating, shown in Supplementary Fig. 9.

**Cell Counting Kit-8.** We first used untreated cells to conduct pre-experiments on the number of cells and the amount of CCK8, and the selection criterion was that the OD value was around 1.0. Subsequently, we seeded the cells into 96-well plates for 10,000 cells per well and added 10ul of CCK8 to each well. Five wells were replicated for each sample. After the 3 h incubation, we measured the absorbance of each well at a wavelength of 450 nm using a microplate reader. After discarding the minimum and maximum values, we used the average for data statistics. Details of the reagents used are in the supplementary Data 1.

**Immunostaining.** All the decidual tissue were fixed in formalin and embedded in paraffin for section. After deparaffinization and hydration, formalin-fixed paraffin embedded decidual sections (4 µm) were subjected to antigen retrieval by autoclaving in 1× sodium citrate solution (pH = 6.0) for 15 min. A diaminobenzidine (ORIGENE) solution was used to visualize antigens. Nuclei were counterstained with hematoxylin. In both IHC and IF studies, paraffin sections were blocked with 5% Bovine Serum Albumin (BSA) in PBS and immune stained by antibodies for JAZF1, Vimentin, BAX, BCL2, cyt c, cleaved-caspase 3, G0S2 and Purβ. Signals were visualized by secondary antibody (1:500, Cell Signaling Technology, 4412) at RT for 1 hr. Details of the reagents used are in the supplementary Data 1.

**ELISA detection.** PRL level was measured in a single-plex assay (Human Prolactin/PRL ELISA Kit, Abcam, CAT# ab226901). All analysis were performed according to the manufacturer's instructions. Absorbance was measured by Bio Tek Instruments ELX808. Details of the reagents used are in the supplementary Data 1.

**RNA extraction and Quantitative real-time PCR (qRT- PCR).** Total RNA was extracted from immortalized HESCs or decidua using RNAiso Plus (TaKaRa, Japan) following the manufacturer's protocol. The quality of RNA was determined by agarose electrophoresis and concentration of RNA was measured by Nanodrop. A total of 1 µg RNA was used to reverse transcribed into cDNA. Quantitative real-time PCR was performed with TB Green (Takara) on an Quantstudio 3 system according to the manufacturer's instructions. The cDNA was predenatured at 95 °C for 30 s, followed by PCR amplification at 95 °C for 5 s and 60 °C for 30 s for 40 cycles. All expression values were normalized with an average CT value of the housekeeping genes *β-actin*. All primers sequences are listed in Supplementary table 6.

**Western blotting analysis.** Western blotting analysis was performed as described previously[52]. Proteins were separated with SDS-PAGE as well as transferred onto polyvinylidene fluoride membranes (PVDF, Millipore, USA). Antibodies against

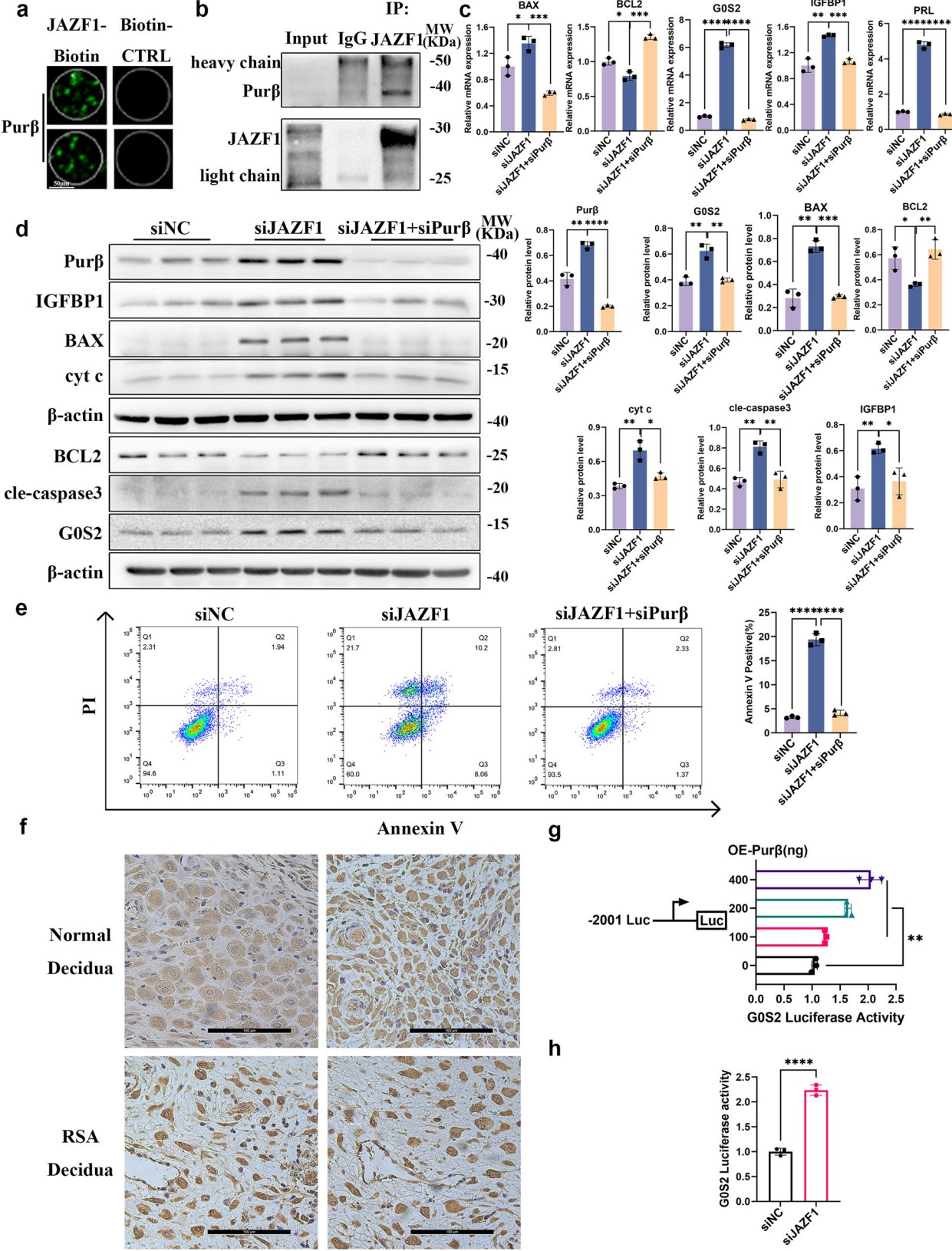

JAZF1, IGFBP1, FOXO1, p-FOXO1, BAX, BCL2, cleaved-caspase3, cyt c, G0S2 and Purβ were used. β-actin served as a loading control. The primary antibodies concentrations were applied according to the manual. The measurement of band intensity of the given protein is represented by the relative protein level value with the loading control β-actin using Image J. Details of the reagents used are in the supplementary Data 1.

**Statistics and reproducibility**. Statistical analysis was performed with GraphPad Prism (v9). The data were shown as the mean ± standard deviation (SD). Statistical analyses were performed using two-tailed unpaired Student's $t$-test or Mann–Whitney $U$-test and among more than two groups by one-way analysis of variance. All experiments were repeated at least three times and $P$-values <0.05 were considered statistically significant.

**Fig. 8 JAZF1 maintains decidua homeostasis by restricting the transcription of G0S2 activated by Purβ in decidualized hESCs. a** Magnified image of Bio-Celastrol binding to Purβ spot on the protein array. scale bars, 50 μm. **b** Co-IP suggested that JAZF1 interacted with Purβ. **c** Results of RT-qPCR showing the expression of *IGFBP1, PRL, BAX, BCL2* and *G0S2* in immortalized HESCs transiently transfected with si*JAZF1* and si*Purβ*. *n* = 3. **d** Protein expression of IGFBP1, G0S2, BCL2, BAX, cleaved-caspase 3, cyt c in immortalized HESCs transfected with si*JAZF1* and si*Purβ* during decidualization. *n* = 3. **e** Flow cytometry of decidualized HESCs transfected si*JAZF1*+si*Purβ*. *n* = 3. **f** IHC analysis of Purβ in RSA decidua compared to normal decidua. *n* = 4. scale bars, 100 μm. **g** Human *G0S2* promoter (-2001 to -1) Luciferase reporter plasmids and Renilla reporter plasmids were transiently co-transfected with *Purβ* expression plasmids into HEK 293 T cells. Luciferase activity was measured 24 h after transfection and normalized to Renilla levels. *n* = 3. **h** Luciferase activity of *G0S2* promoter in decidualized HESCs with co-transfection with OE-*Purβ* and si*JAZF1*. *n* = 3. The data were shown as the mean ± standard deviation (SD). *$P < 0.05$, **$P < 0.01$, ***$P < 0.001$, ****$P < 0.00001$.

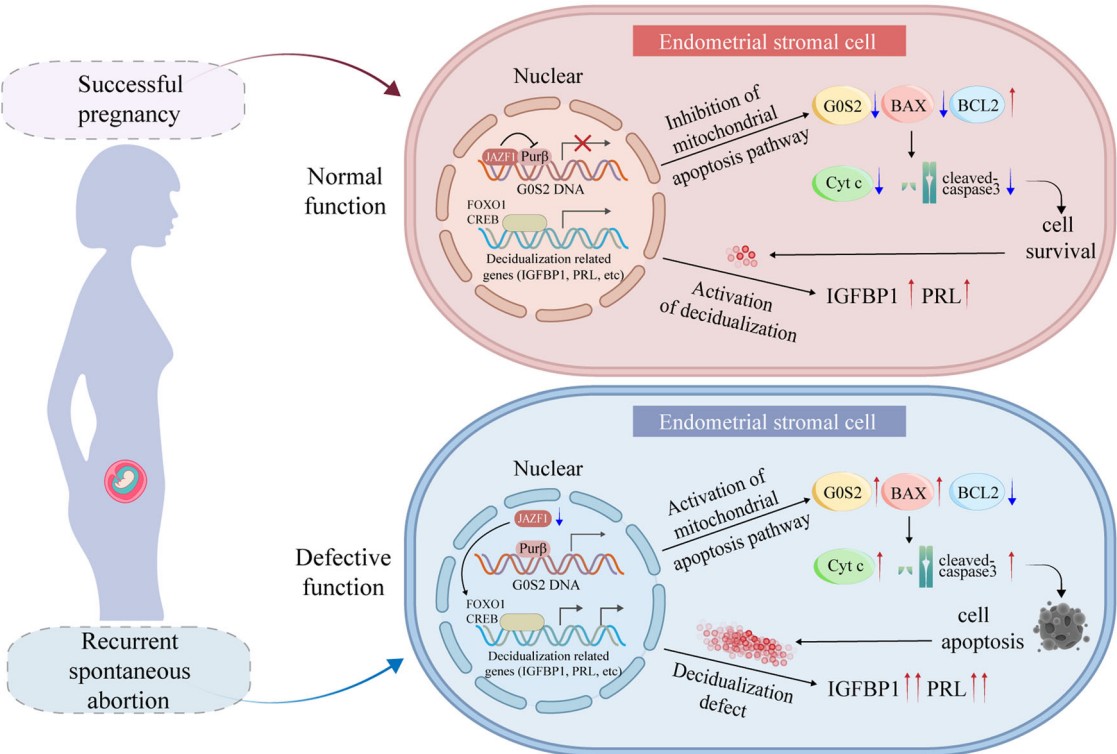

**Fig. 9 Schematic illustration indicating that JAZF1 safeguards hESCs survival and decidualization by repressing the transcription of G0S2 via restricting G0S2 activator Purβ.** JAZF1 depletion promotes the transcriptional activation of G0S2 via enhancing Purβ binding to G0S2 promoter, inducing cell death via apoptosis and decidualization defects in human ESCs of RSA patients. ESCs endometrial stromal cells.

**Reporting summary**. Further information on research design is available in the Nature Portfolio Reporting Summary linked to this article.

## Data availability

The sequencing data generated in this study have been deposited in the Genome Sequence Archive for Human of National Genomics Data Center (https://ngdc.cncb.ac.cn/gsa-human/) database under accession code HRA003581 (RNA-seq), HRA003582 (CUT & Tag in immortalized HESCs). The public datasets used in Fig. 2 and supplementary Fig. 1 were retrieved from microarray and scRNA-seq data deposited in the Gene Expression Omnibus (GDS2052, GSE111976). Uncropped western blot images are available as Supplementary Figs. 10–23. The source data for graphs are available in Supplementary Data 3.

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

## Acknowledgements
This study was supported by grants from the National Key R&D Program of China (No. 2022YFC2704503), the National Natural Science Foundation of China (No. 82071652, 82271695, 81830045 and 82171666), General Program of Guangdong Province Natural Science Foundation (No. 2021A1515011039) and Science and Technology Program of Guangzhou, China (202102010005, 202102010006, and 2023A03J0378).

## Author contributions
L.D., Z.T. and D.C. jointly conceived and supervised the project, interpreted the data and revised the manuscript. Yi.L. participated in bioinformatic analyses, wrote the paper, and prepared figures. Yi.L. and S.L. performed the experiments and collected the data. Yi.L., S.L., L.H., Yu.L., Shanshan Zeng, Y.L., J.L., P.X. and M.L. collected the samples and clinical information. Z.X. performed the pathological examination. Yi.L., L.D., Z.T., Shuang Zhang, J.C. and W.D. participated in data analysis and interpretation. L.D., Z.T., and D.C. edited and finalized manuscript. All authors critically read and commented on the manuscript and approved the final version for submission.

## Competing interests
The authors declare no competing interests.
