## [Peer Review File · Communications Biology]

Reviewers' comments:

Reviewer #1 (Remarks to the Author):

The present manuscript identified and validated JAZF1 was significantly down regulated in stromal cells from RSA decidua. The loss of JAZF1 in hESCs leads to apoptosis and decidualization defects. Meanwhile, the authors present results showing that JAZF1 controls the survival and decidualization of hESCs by limiting G0S2 transcription via restricting the activity of Pur β , which has clinical significance in the pathology of RSA.

Overall, the manuscript sounds interesting but following points need special attention:

Line 126-129, Fig 2A-D, both mRNA and protein expression levels of JAZF1 gradually increased in hESCs accompanied by the up-regulation of decidualization related genes IGFPB1, PRL and FOXO1. However Fig 2A only shows the gene expression level of IGFPB1 and PRL, without the gene expression level of FOXO1. Fig C shows the protein expression level of IGFPB1, JAZF1 and FOXO1, without PRL. Moreover, JAZF1 is implicated in regulating phosphorylation of FOXO1. The phosphorylation level of FOXO1 should be detected.

In addition, the authors uncovered the JAZF1-G0S2-Pur β cascade in the in vitro cells, whether the same scenario exist in vivo should be checked, at least the expression and location of TG0S2 and Pur β in the decidua.

Reviewer #2 (Remarks to the Author):

The manuscript submitted by Liang Y et al., shows that JAZF1 regulates human endometrial stromal cells survival and decidualization by restricting the activity of Pur β , which induces the expression of G0S2, a factor that is involved in the mitochondrial apoptotic pathway in recurrent spontaneous abortion (RSA). The overall manuscript is original, interesting, and scientifically sound. I consider the details provided in the methods section sufficient to reproduce their work. The results are of particular interest in the field of molecular human reproduction and provide novel evidence about specific factors that regulate human decidualization. The overall work is convincing; however, some points should be addressed:

GENERAL

1. I think the results are overestimated in terms of the participation of JAZF1 in stromal decidualization since more studies should be performed to determine its participation in this process, as stated in the manuscript (lines 89-92, 139-140, 370-372, 416-419, 434-436). The relation between apoptosis and decidualization in this study (via JAZF1 and G0S2) is unclear; this is not explained in figure 9. Furthermore, the findings observed in the hESC cell line were not confirmed in primary cultures. They should be confirmed in primary cultures, preferably from both study groups of women included in this study.
2. Authors should discuss the limitation of the sample size used in their study when analyzing RSA samples, considering the heterogeneity and complexity of the disease.
3. Please clarify why the MPA and cAMP cocktail is chosen to induce decidualization. Why not include estradiol? Estradiol is present during the menstrual cycle and pregnancy. Furthermore, estradiol inhibits apoptosis in endometrial stromal cells (<https://doi.org/10.3892/mmr.2018.9428>; <https://www.ijbs.com/v13p0434.htm>; <https://doi.org/10.1093/molehr/gat034>).
4. Authors should include light microscopy images to visualize decidualization-related morphological changes throughout the experiments. In fact, it would be interesting to determine the effect of the several conditions evaluated in this work on the morphology of decidualized stromal cells.
5. Please check minor grammatical errors.

INTRODUCTION

1. The authors should specify when the decidualization occurs (lines 56-57).

2. I suggest including the characteristics of decidual stromal cells and highlighting their differences from the non-decidualized counterpart (line 59).
3. Authors must specify the effect of JAZF1 (increase/decrease?) in the information included in lines 79-81.

RESULTS

1. I suggest including more magnified images in the immunohistochemistry (IHC) figures. The resolution of some IHC and IF images could be improved.
2. Line 133: There is also a protein signal in epithelial cells. The authors should include the negative control (IgG) in fig. 2F.
3. Figure 2: FOXO1 expression (mRNA) results are missing.
4. What is the effect of JAZF1 depletion and overexpression in non-decidualized hESCs?
5. What percentage of dead cells (hESCs) is observed when decidualization is induced in vitro?
6. Line 170: I do not think "restored" is the correct word.
7. Lines 180-181: What about PRL?
8. Lines 182-186 and 423. Could these results be due to the number of viable (or apoptotic) cells obtained by JAZF1 knockdown or overexpression rather than a functional effect?
9. Figure 3 and in the following figures: Did authors obtain the Bax/BCL2 ratio in all conditions? BCL2 is missing in JAZF1 depletion experiments.
10. Line 213: What is the interpretation of this finding? Are primary stromal cells isolated from RSA decidua predisposed to apoptosis even without decidualization stimulus?
11. Lines 298-300, 322-323, and 332: The effect of overexpressing G0S2 and silencing JAZF1 differs when comparing IGFBP1 and PRL expression. Authors should discuss these differences.
12. Fig. 6D: BCL2 and cyt c blots should be improved.
13. The results from Figures 3A and 7A are not consistent (PRL).
14. I suggest including the mass spectrometry results in the supplementary data.

DISCUSSION

1. Authors should briefly discuss the considerations that must be taken into account when comparing the results obtained in decidual tissue and primary cell culture.
2. What is the role of G0S2 in decidualization?
3. Line 407: I missed the part of the manuscript in which the decidualization defect was observed in RSA decidua. I suggest the authors include the data mentioned in line 422; otherwise, there is no clear association between decidualization defect and RSA in terms of the present study.
4. Line 423: the results were not confirmed in primary cell cultures.
5. Lines 451-457: This information summarizes the results. Many points are missing in the discussion.

METHODS

1. Please include exclusion and inclusion criteria.
2. STATISTICS: The authors stated they used statistical analysis to compare two variables; however, there are results with more than two variables.

Reviewer #3 (Remarks to the Author):

A well-written study on how impaired JAZF1, high G0S2, excessive apoptosis and decidualization defect are observed in recurrent spontaneous miscarriage decidua.

In general, the results are well presented in particular the ability to characterize decidual stromal cells from women with RSA. It would have been better, if some of the experiments were performed with primary stromal cells rather than the use of the immortalized stromal cell line. The author need to clearly refer hESCs as "immortalized hESCs" or by the cell line name. Please also comment in the discussion why a cell line was used instead of primary cells.

All amendments are indicated in **blue** font in the revised manuscript. In addition, our point-by-point responses to the latest comments are listed below.

Response to Reviewer(s)' Comments:

Reviewer #1 (Remarks to the Author):

The present manuscript identified and validated JAZF1 was significantly down regulated in stromal cells from RSA decidua. The loss of JAZF1 in hESCs leads to apoptosis and decidualization defects. Meanwhile, the authors present results showing that JAZF1 controls the survival and decidualization of hESCs by limiting G0S2 transcription via restricting the activity of Pur β , which has clinical significance in the pathology of RSA.

Overall, the manuscript sounds interesting but following points need special attention:

(1) Line 126-129, Fig 2A-D, both mRNA and protein expression levels of JAZF1 gradually increased in hESCs accompanied by the up-regulation of decidualization related genes IGFBP1, PRL and FOXO1. However, Fig 2A only shows the gene expression level of IGFBP1 and PRL, without the gene expression level of FOXO1. Fig C shows the protein expression level of IGFBP1, JAZF1 and FOXO1, without PRL.

Response: We appreciate the positive and constructive comments made by the reviewer.

In the revised manuscript, we have changed the figures as suggested. Specifically, we included the gene expression level of FOXO1 in Fig 2A and the PRL protein level of supernatant in Fig 2D.

(2) Moreover, JAZF1 is implicated in regulating phosphorylation of FOXO1. The phosphorylation level of FOXO1 should be detected.

Response: We sincerely appreciate the reviewer's suggestion. To address this concern, we have included the results of FOXO1, p-FOXO1 and p-FOXO1/FOXO1 in Supplementary Fig 2E, G, K, 5B, 6G-H, K and 7C, F-G respectively.

These results suggested that JAZF1 knockdown or G0S2 overexpression could increase the levels of FOXO1 and p-FOXO1 but have no effect on p-FOXO1/FOXO1. As FOXO1 is a key regulator of decidualization¹, the upregulation of FOXO1 upon JAZF1 depletion confirms the role of JAZF1 in decidualization. Previous studies have also shown that overexpression of JAZF1 can promote cardiac microvascular endothelial cell proliferation and angiogenesis via activation of the Akt signaling pathway, and the upregulation of p-FOXO1/FOXO1 was a key node in the activation of this pathway^{2,3}. Our study indicated that JAZF1 knockdown induced apoptosis by activating the mitochondrial apoptosis pathway via upregulation the expression of G0S2, rather than activating the p-FOXO1/FOXO1 pathway.

(3) In addition, the authors uncovered the JAZF1-G0S2-Pur β cascade in the in vitro cells, whether the same scenario exist in vivo should be checked, at least the expression and location of G0S2 and Pur β in the decidua.

Response: We appreciate the reviewer's rigorous comments. we have added the IHC results of Pur β in the decidua in Fig 8F, which showed that Pur β was localized in both

the nucleus and cytoplasm. Moreover, the expression of Pur β in the decidua tissue of the normal group was lower, further suggesting the involvement of Pur β in the transcriptional regulation of G0S2. Additionally, we confirmed the JAZF1-Pur β -G0S2 cascade in two primary endometrial stromal cell lines, and the detailed results were included in Supplementary Fig 1B-D, 2I-M, 4G, 6I-K, and 7E-G. However, the efficiency of G0S2 overexpression in primary endometrial stromal cells was not qualified, and although the efficiency of overexpression of JAZF1 was high, the cells were in poor condition and could not be amplified in the primary culture system.

Reviewer #2 (Remarks to the Author):

The manuscript submitted by Liang Y et al., shows that JAZF1 regulates human endometrial stromal cells survival and decidualization by restricting the activity of Pur β , which induces the expression of G0S2, a factor that is involved in the mitochondrial apoptotic pathway in recurrent spontaneous abortion (RSA). The overall manuscript is original, interesting, and scientifically sound. I consider the details provided in the methods section sufficient to reproduce their work. The results are of particular interest in the field of molecular human reproduction and provide novel evidence about specific factors that regulate human decidualization. The overall work is convincing; however, some points should be addressed:

GENERAL

(1) I think the results are overestimated in terms of the participation of JAZF1 in stromal decidualization since more studies should be performed to determine its participation

in this process, as stated in the manuscript (lines 89-92, 139-140, 370-372, 416-419, 434-436). The relation between apoptosis and decidualization in this study (via JAZF1 and G0S2) is unclear; this is not explained in figure 9.

Response: Thanks for the comment. To unravel the participation of JAZF1 in decidualization, we first detected lower JAZF1 expression in decidual stromal cells of RSA patients. Further large amounts of gene deletion, knockdown and overexpression assays proved the essential roles of JAZF1 in decidualization. We admitted that the relationship between apoptosis caused by JAZF1 loss and decidualization was not clear. To address that, we added a pan-caspase inhibitor (20 μ M Z-VAD-FMK) after JAZF1 knockdown or G0S2 overexpression, then proceeded to decidualization for 6 days to detect decidualization levels. The results indicated that inhibition of caspase can rescue the decidualization defect induced by JAZF1 knockdown or G0S2 overexpression (Supplementary Fig 2L and 5C-E). These results indicate that excessive apoptosis caused by JAZF1 knockdown or G0S2 overexpression leads to decidualization defects. We apologize for any misunderstandings caused by our initial interpretation and thank the reviewer for their insightful comment.

(2) Furthermore, the findings observed in the hESC cell line were not confirmed in primary cultures. They should be confirmed in primary cultures, preferably from both study groups of women included in this study.

Response: We appreciate this thoughtful comment. To confirm our findings in primary cultures, we conducted experiments using primary decidua stromal cells from RSA and

control groups and verified the expression patterns of low JAZF1, high G0S2, and high apoptosis levels observed in decidua. The relevant results were shown in Fig 1F, Fig 4F-G and Fig 5L. Furthermore, we also confirmed the findings observed in hESCs cell line using two primary endometrial stromal cell lines. The relevant results were included in Supplementary Fig 1B-D, 2I-M, 4G, 6I-K, 7E-G, and isolation of primary ESCs was also added to the methods section. However, we were unable to overexpress G0S2 due to the lack of sufficient efficiency, or overexpress JAZF1 because the cells were in poor condition and could not be amplified in the primary culture condition.

(3) Authors should discuss the limitation of the sample size used in their study when analyzing RSA samples, considering the heterogeneity and complexity of the disease.

Response: Thanks for this considerate suggestion. We admit that the number of samples in this paper is limited due to the complexity and heterogeneity of RSA etiology. However, we enrolled patients with two or more previous spontaneous unexplained abortions, normal karyotype of parents and abortus, and absence of uterine malformation, endocrine, metabolic, autoimmune diseases or infection in the RSA group, which ensures the reliability of the results to some extent. In addition, our previous 10× single cell sequencing results verified the JAZF1 protein expression levels in 10 pairs of decidua tissues, among which 8 pairs were consistent with the expression pattern of low JAZF1 in RSA decidua, which supported our conclusion. Nevertheless, we agree that a larger sample size is needed for further confirmation in future studies. We added the discussion of the limitation of sample size in the manuscript, listing as

“In spite of the strengths, there were some limitations that should be paid attention to. Firstly, the sample size in this paper was limited due to the complexity and heterogeneity of RSA etiology. Although we observed consistent results in hESC, primary ESC and decidua tissues possibly due to the strict rules we chose samples, a larger sample scale will further nail down the conclusions we got.”

(4) Please clarify why the MPA and cAMP cocktail is chosen to induce decidualization. Why not include estradiol? Estradiol is present during the menstrual cycle and pregnancy. Furthermore, estradiol inhibits apoptosis in endometrial stromal cells (<https://doi.org/10.3892/mmr.2018.9428>; <https://www.ijbs.com/v13p0434.htm>; <https://doi.org/10.1093/molehr/gat034>).

Response: Thank you for your comment. The MPA and cAMP cocktail is a well-established method and has been used in decidualization induction in many studies *in vitro*⁴⁻⁸. It is reported that the rising progesterone and intracellular cyclic AMP are enough to initiate the decidualization in human endometrium^{9,10}, so we chose MPA and cAMP cocktail only. We agree that estrogen exists in the menstrual cycle and pregnancy, and low concentration of estrogen can promote the proliferation of endometrial stromal cells and inhibit early apoptosis¹¹, but not late apoptosis or cell death. The purpose of this study was to demonstrate that JAZF1 knockdown induced cell death via apoptosis of decidualized hESCs. As shown in Fig 3H, the number of PI⁺/Annexin V⁺ cells increased the most after JAZF1 knockdown, called late apoptotic or dead cells. It may be that apoptosis has been used to represent the change of phenotype in our paper, which

is not an accurate description of the changes in cells. We rephrased our text to avoid any ambiguity, listing as “cell death through mitochondrial apoptosis pathway”.

(5) Authors should include light microscopy images to visualize decidualization-related morphological changes throughout the experiments. In fact, it would be interesting to determine the effect of the several conditions evaluated in this work on the morphology of decidualized stromal cells.

Response: We appreciate the reviewer’s rigorous comments. The light microscopy images of decidualized stromal cells have been included in Supplementary Fig 2F. The results also proved that JAZF1 knockdown led to decidualization defect and decreased survival rate of stromal cells, while JAZF1 overexpression showed the opposite way.

(6) Please check minor grammatical errors.

Response: Thanks for pointing this out. We have made changes to the relevant grammatical errors.

INTRODUCTION

(1) The authors should specify when the decidualization occurs (lines 56-57).

Response: Thanks for the suggestions. The relevant information has been added to the second paragraph of the introduction showing as “During pregnancy establishment and maintenance, the hESCs undergo dynamic transformation during the progesterone-dominant early secretory phase, which is called decidualization”.

(2) I suggest including the characteristics of decidual stromal cells and highlighting their differences from the non-decidualized counterpart (line 59).

Response: Thanks for the comments. The characteristics of decidualized cells have been added to the second paragraph of the introduction, listing as “The hallmark of decidualization is the differentiation of fibroblast-like hESCs into decidual stromal cells (DSCs) characterized by rounding of the nucleus, increased number of nucleoli, dilatation of the rough endoplasmic reticulum (rER) and Golgi systems”.

(3) Authors must specify the effect of JAZF1 (increase/decrease?) in the information included in lines 79-81.

Response: Thanks to the reviewers for the comment. We have revised the relevant content as “Furthermore, overexpression of JAZF1 is also implicated in downregulating phosphorylation of FKHR (also known as forkhead box O1 (FOXO1)) and cAMP response element-binding protein (CREB), which are key regulators of decidualization1.”.

RESULTS

(1) I suggest including more magnified images in the immunohistochemistry (IHC) figures. The resolution of some IHC and IF images could be improved.

Response: We appreciate this constructive suggestion. Higher resolution and magnified images of IHC and IF results have been added.

(2) Line 133: There is also a protein signal in epithelial cells. The authors should include the negative control (IgG) in fig. 2F.

Response: We appreciate the reviewer's rigorous comments. The IHC results of negative control (IgG) were included in Fig 2G.

(3) Figure 2: FOXO1 expression (mRNA) results are missing.

Response: This critical comment of the reviewer is highly appreciated. FOXO1 gene expression (mRNA) was provided in Fig 2A.

(4) What is the effect of JAZF1 depletion and overexpression in non-decidualized hESCs?

Response: We thank the reviewer for this comment. As this study mainly focuses on the role of JAZF1 in decidualization, the results of JAZF1 depletion and overexpression in non-decidualized hESCs were not presented in the manuscript. However, we did perform these experiments. In non-decidualized hESCs, JAZF1 knockdown resulted in slower growth and more dead cells, while JAZF1 overexpression was not significant. The apoptosis of non-decidualized hESCs after knockdown and overexpression of JAZF1 were also detected by flow cytometry. The results showed that apoptosis was upregulated after JAZF1 knockdown, but not significant in JAZF1 overexpression. The results were not included in the manuscript and were shown as below.

(5) What percentage of dead cells (hESCs) is observed when decidualization is induced in vitro?

Response: Thank you for the question. We observed that the cell death rate of decidualized HESCs transfected with siNC was about 5.5%, while the percentage of siJAZF1 was about 12.3%, suggesting that JAZF1 down-regulation induced cell death.

The results were shown as below.

(6) Line 170: I do not think “restored” is the correct word.

Response: Thanks for the comment. We rephrased our text to avoid any ambiguity.

“Restored” is changed to “downregulated”.

(7) Lines 180-181: What about PRL?

Response: We sincerely appreciate the reviewer's valuable comment. PRL, reflected by both mRNA and protein expression, has been demonstrated to be present only in late-luteal phase endometrium, either with or without the presence of a pregnancy, as well as during subsequent stages of pregnancy¹². Robbert et al. demonstrated premature decidualization during the luteal phase, reflected by PRL expression, may lead to embryo-endometrial asynchrony, and consequently RIF¹³. Furthermore, hyperprolactinemia may contribute to infertility by inducing decreased GnRH production, inhibition of corpus luteum function, endometrial dysfunction and decreased implantation¹⁴. Panzan et al.¹⁵ indicated MCP-induced hyperprolactinemia negatively affects ovarian function, endometrial morphology and embryo implantation in mice. Thus, a certain level of circulating PRL also might be necessary for optimal reproductive outcomes.

(8) Lines 182-186 and 423. Could these results be due to the number of viable (or apoptotic) cells obtained by JAZF1 knockdown or overexpression rather than a functional effect?

Response: We thank the reviewer for this insightful question. We admit that JAZF1 knockdown may result in a decrease in the number of stromal cells, which could potentially affect the result of transwell assay. Therefore, we also set up transwell experiments for JAZF1-overexpressed hESCs. Details of the experiment are as follows. Equal numbers of OE-NC and OE-JAZF1 cells were transplanted into 6-well plates and

decidualized for 6 days. The supernatant was collected for subsequent transwell experiment. At the same time, there was no significant difference in apoptosis between JAZF1-overexpression cells and the control group, that is, the number of cells undergoing transwell was not much different. Therefore, in this experiment, JAZF1-overexpression hESCs promoted HTR-8 invasion because of cell functional effect, rather than changes in cell number.

(9) Figure 3 and in the following figures: Did authors obtain the Bax/BCL2 ratio in all conditions? BCL2 is missing in JAZF1 depletion experiments.

Response: Thanks for the comment. The BAX/BCL2 ratio has been added to the supplementary figures corresponding to each figure and BCL2 expression was added to Fig 3D.

(10) Line 213: What is the interpretation of this finding? Are primary stromal cells isolated from RSA decidua predisposed to apoptosis even without decidualization stimulus?

Response: The critical comment of the reviewer is highly appreciated. We apologize for any confusion caused. This result mainly indicated that excessive apoptosis existed in primary stromal cells from RSA decidua with decidualization, as evidenced by the extraction of total RNA and protein after 6 hours of adherence to the dishes from primary decidua stromal cells without *in vitro* culture. Therefore, this result may indicate a higher level of apoptosis in RSA decidua stromal cells undergoing

decidualization.

(11) Lines 298-300, 322-323, and 332: The effect of overexpressing G0S2 and silencing JAZF1 differs when comparing IGFBP1 and PRL expression. Authors should discuss these differences.

Response: Thank you for the suggestion. In this study, we found that JAZF1 knockdown alone or G0S2 overexpression alone could cause decidualization defects, and affect the expression of both IGFBP1 and PRL, indicating that JAZF1 and G0S2 are involved in the regulation of decidualization. However, when JAZF1 and G0S2 were co-downregulated, they only down-regulated IGFBP1 expression, but not PRL expression, which suggests that PRL may be regulated by other downstream networks of JAZF1. We have discussed this result in the discussion section, listing as “Previous studies have found that G0S2 induces cell apoptosis by binding to BCL2 and activating mitochondrial apoptosis pathway. In the current study, we found that G0S2 was a novel molecule that regulated stromal cells survival and decidualization. Mechanistic studies led to the discovery that JAZF1 depletion induced cell death via mitochondrial apoptosis in stromal cells through activation of G0S2. However, when JAZF1 and G0S2 were co-downregulated, they only down-regulated IGFBP1 expression, but not PRL expression, which suggests that PRL may be regulated by other downstream networks of JAZF1.”.

(12) Fig. 6D: BCL2 and cyt c blots should be improved.

Response: Thanks for the suggestion. We repeated the experiment and replaced the original figure.

(13) The results from Figures 3A and 7A are not consistent (PRL).

Response: Thanks for your concern. It may be caused by technical issues of sampling when running qRT-PCR. To further confirm the data, we have repeated the experiment and updated the results.

(14) I suggest including the mass spectrometry results in the supplementary data.

Response: Thanks for the advice. We have attached the mass spectrometry results to Supplementary Table 8.

DISCUSSION

(1) Authors should briefly discuss the considerations that must be taken into account when comparing the results obtained in decidual tissue and primary cell culture.

Response: The critical comment of the reviewer is highly appreciated. Comparing the results obtained in decidual tissue and primary cell culture requires careful consideration of several factors. Primary cell culture is a simplified system that only includes a single cell type, while the decidual tissue is composed of multiple cell types that may interact and contribute to the decidualization process. Therefore, it is essential to keep in mind that the contribution of other cells to decidualization cannot be excluded. Further studies are needed to explore the role of cell interaction in decidualization. We

have added a discussion of these considerations to the penultimate paragraph of the manuscript. The discussion is shown as “In addition, decidualization involves multiple cell types in decidual tissue, so the contribution of other cells to decidualization cannot be excluded. In this study, we focused mainly on stromal cells alone. Further studies are needed to explore the role of cell interaction in decidualization.”.

(2) What is the role of G0S2 in decidualization?

Response: We appreciate the reviewer’s rigorous comments. We apologize for not making this clear. We found that overexpression of G0S2 induced decidualization defects in stromal cells, which were specifically manifested as abnormal expression of IGFBP1, PRL, and FOXO1, suggesting that G0S2 may be involved in the regulation of decidualization and induction of decidualization defects. Moreover, in the rescue experiment, co-knockdown of JAZF1 and G0S2 partially restored the high level of IGFBP1, but had no effect on PRL and FOXO1 expression, indicating that the JAZF1-G0S2 network is involved in regulating the expression of IGFBP1 but not the regulatory pathway of PRL and FOXO1. We have added this clarification to the manuscript.

(3) Line 407: I missed the part of the manuscript in which the decidualization defect was observed in RSA decidua. I suggest the authors include the data mentioned in line 422; otherwise, there is no clear association between decidualization defect and RSA in terms of the present study.

Response: The critical comment of the reviewer is highly appreciated. We apologize

for not making this clear. The results of decidualization defect observed in RSA decidua have been requoted from our published data¹⁶. The results are as follows.

Fig. 3. Decidualization markers were also increased in RPL decidua tissues. (A-B) IGFBP1 protein level in RPL decidua tissues and control samples by IHC via tissue microarray (A) (n = 9) and western blot (B) (n = 20); (C-D) IGFBP1 mRNA level in RPL decidua tissues and control samples by in situ hybridization(C) and ScRNA-seq analysis in DS1 cluster; (E-F) qRT-PCR(E) and ELISA detection(F) showed the PRL expression level in RPL decidua tissues and control samples (E: control=23, RPL=22; F: control=11, RPL=12). ** indicates P < 0.01. **** indicates P < 0.0001.

(4) Line 423: the results were not confirmed in primary cell cultures.

Response: We sincerely appreciate the reviewer's important comment. We have replicated the results of the immortalized cell lines using the primary endometrial stromal cells, and the results were consistent (Supplementary Fig 2L).

(5) Lines 451-457: This information summarizes the results. Many points are missing in the discussion.

Response: We appreciate the reviewer's rigorous comments. We rephrased the

discussion as “Our results proved that JAZF1 worked as a typical transcription repressor just like in other systems. However, JAZF1 targeted apoptosis-related gene G0S2 in immortalized HESCs during decidualization, implying cell type specific functions of JAZF1. Moreover, as a transcription repressor, whether JAZF1 directly regulates the transcription of key decidualization-related genes requires further experiments.” In addition, we have added the role of G0S2 and Pur β in decidualization in the discussion section, listing as “Previous studies have found that G0S2 induces cell apoptosis by binding to BCL2 and activating mitochondrial apoptosis pathway. In the current study, we found that G0S2 was a novel molecule that regulated stromal cells survival and decidualization. Mechanistic studies led to the discovery that JAZF1 depletion induced cell death via mitochondrial apoptosis in stromal cells through activation of G0S2. However, when JAZF1 and G0S2 were co-downregulated, they only down-regulated IGFBP1 expression, but not PRL expression, which suggests that PRL may be regulated by other downstream networks of JAZF1. Transcription factor Pur β has been found to bind to DNA promoter sequences of several genes and thus participate in transcriptional regulation, but the effect of its expression on cell function has not been described. This study showed that Pur β could bind to G0S2 promoter and promote its transcription, thus inducing cell death and decidualization defects. More importantly, knockdown of Pur β or G0S2 restored the apoptotic defects of JAZF1-knockdown HESCs or primary ESCs”.

METHODS

(1) Please include exclusion and inclusion criteria.

Response: Thanks for the comments. We have provided the exclusion and inclusion criteria as following in the part of “Sample of clinical cases” in methods. The methods is shown as “Patients with endocrine disorders, metabolic, or autoimmune diseases were excluded. Only patients with unexplained RSA were recruited in the study”.

(2) STATISTICS: The authors stated they used statistical analysis to compare two variables; however, there are results with more than two variables.

Response: We sincerely appreciate the reviewer’s important comment. Statistical analyses were performed using two-tailed unpaired Student’s t-test or Mann-Whitney U-test for comparisons between two groups and one-way analysis of variance (ANOVA) for comparisons among more than two groups. We have revised our text to clarify the statistical methods used in the study.

Reviewer #3 (Remarks to the Author):

A well-written study on how impaired JAZF1, high G0S2, excessive apoptosis and decidualization defect are observed in recurrent spontaneous miscarriage decidua. In general, the results are well presented in particular the ability to characterize decidual stromal cells from women with RSA. It would have been better, if some of the experiments were performed with primary stromal cells rather than the use of the immortalized stromal cell line. The author needs to clearly refer hESCs as "immortalized hESCs" or by the cell line name. Please also comment in the discussion

why a cell line was used instead of primary cells.

Response: We sincerely appreciate the reviewer's valuable comments. We have clarified in the manuscript that the hESCs used in the study are immortalized hESCs. We agree with the reviewer that it would have been better to include experiments with primary stromal cells. As suggested, we have now included experiments using primary endometrial stromal cells, and the results are consistent with those obtained using the immortalized cell line. We have added the relevant results to Supplementary Fig 1B-D, 2I-M, 4G, 6I-K, and 7E-G. However, the efficiency of G0S2 overexpression in primary endometrial stromal cells was not satisfactory, and although JAZF1 overexpression efficiency was high, the cells were in poor condition and could not be amplified, which prevented us from repeating the relevant experiments in the primary culture system.

- 1 Vashistha, A., Khan, H. R. & Rudraiah, M. Role of cAMP/PKA/CREB pathway and β -arrestin 1 in LH induced luteolysis in pregnant rats. *Reproduction* **162**, 21-31 (2021). <https://doi.org:10.1530/REP-20-0661>
- 2 Huang, F. *et al.* FoxO1-mediated inhibition of STAT1 alleviates tubulointerstitial fibrosis and tubule apoptosis in diabetic kidney disease. *EBioMedicine* **48**, 491-504 (2019). <https://doi.org:10.1016/j.ebiom.2019.09.002>
- 3 Shang, J., Gao, Z.-Y., Zhang, L.-Y. & Wang, C.-Y. Over-expression of JAZF1 promotes cardiac microvascular endothelial cell proliferation and angiogenesis via activation of the Akt signaling pathway in rats with myocardial ischemia-reperfusion. *Cell Cycle* **18**, 1619-1634 (2019). <https://doi.org:10.1080/15384101.2019.1629774>
- 4 Zhou, Q. *et al.* EHD1 impairs decidualization by regulating the Wnt4/ β -catenin signaling pathway in recurrent implantation failure. *EBioMedicine* **50**, 343-354 (2019). <https://doi.org:10.1016/j.ebiom.2019.10.018>
- 5 Wang, Z. *et al.* ATF3 deficiency impairs the proliferative-secretory phase transition and decidualization in RIF patients. *Cell Death Dis* **12**, 387 (2021). <https://doi.org:10.1038/s41419-021-03679-8>
- 6 Rawlings, T. M. *et al.* Modelling the impact of decidual senescence on embryo implantation in human endometrial assembloids. *Elife* **10** (2021). <https://doi.org:10.7554/eLife.69603>

- 7 Yao, S. *et al.* Resveratrol alleviates zea-induced decidualization disturbance
in human endometrial stromal cells. *Ecotoxicol Environ Saf* **207**, 111511 (2021).
<https://doi.org/10.1016/j.ecoenv.2020.111511>
- 8 Lucas, E. S. *et al.* Recurrent pregnancy loss is associated with a pro-
senescent decidual response during the peri-implantation window. *Commun Biol*
3, 37 (2020). <https://doi.org/10.1038/s42003-020-0763-1>
- 9 Gellersen, B. & Brosens, J. J. Cyclic decidualization of the human endometrium
in reproductive health and failure. *Endocr Rev* **35**, 851–905 (2014).
<https://doi.org/10.1210/er.2014-1045>
- 10 Cha, J., Sun, X. & Dey, S. K. Mechanisms of implantation: strategies for
successful pregnancy. *Nat Med* **18**, 1754–1767 (2012).
<https://doi.org/10.1038/nm.3012>
- 11 Shao, J. *et al.* Estrogen promotes the growth of decidual stromal cells in
human early pregnancy. *Mol Hum Reprod* **19**, 655–664 (2013).
<https://doi.org/10.1093/molehr/gat034>
- 12 Altmäe, S. *et al.* Meta-signature of human endometrial receptivity: a meta-
analysis and validation study of transcriptomic biomarkers. *Sci Rep* **7**, 10077
(2017). <https://doi.org/10.1038/s41598-017-10098-3>
- 13 Berkhout, R. P., Lambalk, C. B., Repping, S., Hamer, G. & Mastenbroek, S.
Premature expression of the decidualization marker prolactin is associated
with repeated implantation failure. *Gynecol Endocrinol* **36**, 360–364 (2020).
<https://doi.org/10.1080/09513590.2019.1650344>
- 14 Iancu, M. E., Albu, A. I. & Albu, D. N. Prolactin Relationship with Fertility
and In Vitro Fertilization Outcomes—A Review of the Literature.
Pharmaceuticals (Basel) **16** (2023). <https://doi.org/10.3390/ph16010122>
- 15 Panzan, M. Q. *et al.* Metoclopramide-induced hyperprolactinaemia caused marked
decline in pinopodes and pregnancy rates in mice. *Hum Reprod* **21**, 2514–2520
(2006).
- 16 Zeng, S. *et al.* TNF α /TNFR1 signal induces excessive senescence of decidual
stromal cells in recurrent pregnancy loss. *J Reprod Immunol* **155**, 103776 (2023).
<https://doi.org/10.1016/j.jri.2022.103776>

REVIEWERS' COMMENTS:

Reviewer #1 (Remarks to the Author):

The submission has been greatly improved and is worthy of publication.

Reviewer #2 (Remarks to the Author):

I did not find supplementary table 8. I suggest the authors indicate in the "Response to Reviewer(s)' Comments" letter where they made the modifications in the new version of the manuscript (i.e. line xxx). I have no further comments.

Reviewer #3 (Remarks to the Author):

No further comments

All amendments are indicated in **blue** font in the revised manuscript. In addition, our point-by-point responses to the latest comments are listed below.

Response to Reviewer(s)' Comments:

Reviewer #2 (Remarks to the Author):

1. I did not find supplementary table 8.

Response: Thanks for pointing this out. The file of Supplementary table 8, named Supplementary Data 2, has been uploaded to the system.

2. I suggest the authors indicate in the "Response to Reviewer(s)' Comments" letter where they made the modifications in the new version of the manuscript (i.e. line xxx).

I have no further comments.

Response: Thanks for this considerate suggestion. According to the latest uploaded file, we added the corresponding line number of the specific modified content to the previous review comments. The details are attached below.

The manuscript submitted by Liang Y et al., shows that JAZF1 regulates human endometrial stromal cells survival and decidualization by restricting the activity of Pur β , which induces the expression of G0S2, a factor that is involved in the mitochondrial apoptotic pathway in recurrent spontaneous abortion (RSA). The overall manuscript is original, interesting, and scientifically sound. I consider the details provided in the

methods section sufficient to reproduce their work. The results are of particular interest in the field of molecular human reproduction and provide novel evidence about specific factors that regulate human decidualization. The overall work is convincing; however, some points should be addressed:

GENERAL

(1) I think the results are overestimated in terms of the participation of JAZF1 in stromal decidualization since more studies should be performed to determine its participation in this process, as stated in the manuscript (lines 89-92, 139-140, 370-372, 416-419, 434-436). The relation between apoptosis and decidualization in this study (via JAZF1 and G0S2) is unclear; this is not explained in figure 9.

Response: Thanks for the comment. To unravel the participation of JAZF1 in decidualization, we first detected lower JAZF1 expression in decidual stromal cells of RSA patients. Further large amounts of gene deletion, knockdown and overexpression assays proved the essential roles of JAZF1 in decidualization. We admitted that the relationship between apoptosis caused by JAZF1 loss and decidualization was not clear. To address that, we added a pan-caspase inhibitor (20 μ M Z-VAD-FMK) after JAZF1 knockdown or G0S2 overexpression, then proceeded to decidualization for 6 days to detect decidualization levels. The results indicated that inhibition of caspase can rescue the decidualization defect induced by JAZF1 knockdown or G0S2 overexpression (line:161-165, Supplementary Fig 2l-m and line:251-254, Supplementary 6c-e). These results indicate that excessive apoptosis caused by JAZF1 knockdown or G0S2 overexpression leads to decidualization defects. We apologize for any

misunderstandings caused by our initial interpretation and thank the reviewer for their insightful comment.

(2) Furthermore, the findings observed in the hESC cell line were not confirmed in primary cultures. They should be confirmed in primary cultures, preferably from both study groups of women included in this study.

Response: We appreciate this thoughtful comment. To confirm our findings in primary cultures, we conducted experiments using primary decidua stromal cells from RSA and control groups and verified the expression patterns of low JAZF1, high G0S2, and high apoptosis levels observed in decidua. The relevant results were shown in line:103-104, 231-232, 186-189. Furthermore, we also confirmed the findings observed in hESCs cell line using two primary endometrial stromal cell lines. The relevant results were included in line:115-116, 154-155, 227-228, 270-271, 295-296, and isolation of primary ESCs was also added to the methods section. However, we were unable to overexpress G0S2 due to the lack of sufficient efficiency, or overexpress JAZF1 because the cells were in poor condition and could not be amplified in the primary culture condition.

(3) Authors should discuss the limitation of the sample size used in their study when analyzing RSA samples, considering the heterogeneity and complexity of the disease.

Response: Thanks for this considerate suggestion. We admit that the number of samples in this paper is limited due to the complexity and heterogeneity of RSA etiology. However, we enrolled patients with two or more previous spontaneous unexplained

abortions, normal karyotype of parents and abortus, and absence of uterine malformation, endocrine, metabolic, autoimmune diseases or infection in the RSA group, which ensures the reliability of the results to some extent. In addition, our previous 10× single cell sequencing results verified the JAZF1 protein expression levels in 10 pairs of decidua tissues, among which 8 pairs were consistent with the expression pattern of low JAZF1 in RSA decidua, which supported our conclusion. Nevertheless, we agree that a larger sample size is needed for further confirmation in future studies. We added the discussion of the limitation of sample size in the manuscript, listing as “line:391-395. In spite of the strengths, there were some limitations that should be paid attention to. Firstly, the sample size in this paper was limited due to the complexity and heterogeneity of RSA etiology. Although we observed consistent results in hESC, primary ESC and decidua tissues possibly due to the strict rules we chose samples, a larger sample scale will further nail down the conclusions we got.”

(4) Please clarify why the MPA and cAMP cocktail is chosen to induce decidualization. Why not include estradiol? Estradiol is present during the menstrual cycle and pregnancy. Furthermore, estradiol inhibits apoptosis in endometrial stromal cells (<https://doi.org/10.3892/mmr.2018.9428>; <https://www.ijbs.com/v13p0434.htm>; <https://doi.org/10.1093/molehr/gat034>).

Response: Thank you for your comment. The MPA and cAMP cocktail is a well-established method and has been used in decidualization induction in many studies *in vitro*⁴⁻⁸. It is reported that the rising progesterone and intracellular cyclic AMP are

enough to initiate the decidualization in human endometrium^{9,10}, so we chose MPA and cAMP cocktail only. We agree that estrogen exists in the menstrual cycle and pregnancy, and low concentration of estrogen can promote the proliferation of endometrial stromal cells and inhibit early apoptosis¹¹, but not late apoptosis or cell death. The purpose of this study was to demonstrate that JAZF1 knockdown induced cell death via apoptosis of decidualized hESCs. As shown in Fig 3h, the number of PI⁺/Annexin V⁺ cells increased the most after JAZF1 knockdown, called late apoptotic or dead cells. It may be that apoptosis has been used to represent the change of phenotype in our paper, which is not an accurate description of the changes in cells. We rephrased our text to avoid any ambiguity, listing as “cell death through mitochondrial apoptosis pathway”.

(5) Authors should include light microscopy images to visualize decidualization-related morphological changes throughout the experiments. In fact, it would be interesting to determine the effect of the several conditions evaluated in this work on the morphology of decidualized stromal cells.

Response: We appreciate the reviewer’s rigorous comments. The light microscopy images of decidualized stromal cells have been included in line:147-149. The results also proved that JAZF1 knockdown led to decidualization defect and decreased survival rate of stromal cells, while JAZF1 overexpression showed the opposite way.

(6) Please check minor grammatical errors.

Response: Thanks for pointing this out. We have made changes to the relevant

grammatical errors.

INTRODUCTION

(1) The authors should specify when the decidualization occurs (lines 56-57).

Response: Thanks for the suggestions. The relevant information has been added to the second paragraph of the introduction showing as “line:53-55. During pregnancy establishment and maintenance, the hESCs undergo dynamic transformation during the progesterone-dominant early secretory phase, which is called decidualization”.

(2) I suggest including the characteristics of decidual stromal cells and highlighting their differences from the non-decidualized counterpart (line 59).

Response: Thanks for the comments. The characteristics of decidualized cells have been added to the second paragraph of the introduction, listing as “line:55-58. The hallmark of decidualization is the differentiation of fibroblast-like hESCs into decidual stromal cells (DSCs) characterized by rounding of the nucleus, increased number of nucleoli, dilatation of the rough endoplasmic reticulum (rER) and Golgi systems”.

(3) Authors must specify the effect of JAZF1 (increase/decrease?) in the information included in lines 79-81.

Response: Thanks to the reviewers for the comment. We have revised the relevant content as “line:77-78. Furthermore, overexpression of JAZF1 is also implicated in downregulating phosphorylation of FKHR (also known as forkhead box O1 (FOXO1))

and cAMP response element-binding protein (CREB), which are key regulators of decidualization1.”.

RESULTS

(1) I suggest including more magnified images in the immunohistochemistry (IHC) figures. The resolution of some IHC and IF images could be improved.

Response: We appreciate this constructive suggestion. Higher resolution and magnified images of IHC and IF results have been added.

(2) Line 133: There is also a protein signal in epithelial cells. The authors should include the negative control (IgG) in fig. 2F.

Response: We appreciate the reviewer’s rigorous comments. The IHC results of negative control (IgG) were included in Fig 2G.

(3) Figure 2: FOXO1 expression (mRNA) results are missing.

Response: This critical comment of the reviewer is highly appreciated. FOXO1 gene expression (mRNA) was provided in line:112-115.

(4) What is the effect of JAZF1 depletion and overexpression in non-decidualized hESCs?

Response: We thank the reviewer for this comment. As this study mainly focuses on the role of JAZF1 in decidualization, the results of JAZF1 depletion and overexpression in

non-decidualized hESCs were not presented in the manuscript. However, we did perform these experiments. In non-decidualized hESCs, JAZF1 knockdown resulted in slower growth and more dead cells, while JAZF1 overexpression was not significant. The apoptosis of non-decidualized hESCs after knockdown and overexpression of JAZF1 were also detected by flow cytometry. The results showed that apoptosis was upregulated after JAZF1 knockdown, but not significant in JAZF1 overexpression. The results were not included in the manuscript and were shown as below.

(5) What percentage of dead cells (hESCs) is observed when decidualization is induced in vitro?

Response: Thank you for the question. We observed that the cell death rate of decidualized HESCs transfected with siNC was about 5.5%, while the percentage of siJAZF1 was about 12.3%, suggesting that JAZF1 down-regulation induced cell death. The results were shown as below.

(6) Line 170: I do not think “restored” is the correct word.

Response: Thanks for the comment. We rephrased our text to avoid any ambiguity.

“Restored” is changed to “line:150, downregulated”.

(7) Lines 180-181: What about PRL?

Response: We sincerely appreciate the reviewer’s valuable comment. PRL, reflected by both mRNA and protein expression, has been demonstrated to be present only in late-luteal phase endometrium, either with or without the presence of a pregnancy, as well as during subsequent stages of pregnancy¹². Line:170-171. Robbert et al. demonstrated premature decidualization during the luteal phase, reflected by PRL expression, may lead to embryo-endometrial asynchrony, and consequently RIF¹³. Furthermore, hyperprolactinemia may contribute to infertility by inducing decreased GnRH production, inhibition of corpus luteum function, endometrial dysfunction and decreased implantation¹⁴. Panzan et al.¹⁵ indicated MCP-induced hyperprolactinemia negatively affects ovarian function, endometrial morphology and embryo implantation in mice. Thus, a certain level of circulating PRL also might be necessary for optimal reproductive outcomes.

(8) Lines 182-186 and 423. Could these results be due to the number of viable (or apoptotic) cells obtained by JAZF1 knockdown or overexpression rather than a functional effect?

Response: We thank the reviewer for this insightful question. We admit that JAZF1 knockdown may result in a decrease in the number of stromal cells, which could potentially affect the result of transwell assay. Therefore, we also set up transwell experiments for JAZF1-overexpressed hESCs. Details of the experiment are as follows. Equal numbers of OE-NC and OE-JAZF1 cells were transplanted into 6-well plates and decidualized for 6 days. The supernatant was collected for subsequent transwell experiment. At the same time, there was no significant difference in apoptosis between JAZF1-overexpression cells and the control group, that is, the number of cells undergoing transwell was not much different. Therefore, in this experiment, JAZF1-overexpression hESCs promoted HTR-8 invasion because of cell functional effect, rather than changes in cell number.

(9) Figure 3 and in the following figures: Did authors obtain the Bax/BCL2 ratio in all conditions? BCL2 is missing in JAZF1 depletion experiments.

Response: Thanks for the comment. The BAX/BCL2 ratio has been added to the supplementary figures corresponding to each figure and BCL2 expression was added in line:142-143,150,187,244,263,290.

(10) Line 213: What is the interpretation of this finding? Are primary stromal cells isolated from RSA decidua predisposed to apoptosis even without decidualization stimulus?

Response: The critical comment of the reviewer is highly appreciated. We apologize for any confusion caused. This result mainly indicated that excessive apoptosis existed in primary stromal cells from RSA decidua with decidualization, as evidenced by the extraction of total RNA and protein after 6 hours of adherence to the dishes from primary decidua stromal cells without *in vitro* culture. Therefore, this result may indicate a higher level of apoptosis in RSA decidua stromal cells undergoing decidualization.

(11) Lines 298-300, 322-323, and 332: The effect of overexpressing G0S2 and silencing JAZF1 differs when comparing IGFBP1 and PRL expression. Authors should discuss these differences.

Response: Thank you for the suggestion. In this study, we found that JAZF1 knockdown alone or G0S2 overexpression alone could cause decidualization defects, and affect the expression of both IGFBP1 and PRL, indicating that JAZF1 and G0S2 are involved in the regulation of decidualization. However, when JAZF1 and G0S2 were co-downregulated, they only down-regulated IGFBP1 expression, but not PRL expression, which suggests that PRL may be regulated by other downstream networks of JAZF1. We have discussed this result in the discussion section, listing as “line:378-385.

Previous studies have found that G0S2 induces cell apoptosis by binding to BCL2 and

activating mitochondrial apoptosis pathway. In the current study, we found that G0S2 was a novel molecule that regulated stromal cells survival and decidualization. Mechanistic studies led to the discovery that JAZF1 depletion induced cell death via mitochondrial apoptosis in stromal cells through activation of G0S2. However, when JAZF1 and G0S2 were co-downregulated, they only down-regulated IGFBP1 expression, but not PRL expression, which suggests that PRL may be regulated by other downstream networks of JAZF1.”.

(12) Fig. 6D: BCL2 and cyt c blots should be improved.

Response: Thanks for the suggestion. We repeated the experiment and replaced the original figure.

(13) The results from Figures 3A and 7A are not consistent (PRL).

Response: Thanks for your concern. It may be caused by technical issues of sampling when running qRT-PCR. To further confirm the data, we have repeated the experiment and updated the results.

(14) I suggest including the mass spectrometry results in the supplementary data.

Response: Thanks for the advice. We have attached the mass spectrometry results to Supplementary Table 8 (file is named Supplementary Data 2).

DISCUSSION

(1) Authors should briefly discuss the considerations that must be taken into account when comparing the results obtained in decidual tissue and primary cell culture.

Response: The critical comment of the reviewer is highly appreciated. Comparing the results obtained in decidual tissue and primary cell culture requires careful consideration of several factors. Primary cell culture is a simplified system that only includes a single cell type, while the decidual tissue is composed of multiple cell types that may interact and contribute to the decidualization process. Therefore, it is essential to keep in mind that the contribution of other cells to decidualization cannot be excluded. Further studies are needed to explore the role of cell interaction in decidualization. We have added a discussion of these considerations to the penultimate paragraph of the manuscript. The discussion is shown as “line:395-399. In addition, decidualization involves multiple cell types in decidual tissue, so the contribution of other cells to decidualization cannot be excluded. In this study, we focused mainly on stromal cells alone. Further studies are needed to explore the role of cell interaction in decidualization.”.

(2) What is the role of G0S2 in decidualization?

Response: We appreciate the reviewer’s rigorous comments. We apologize for not making this clear. We found that overexpression of G0S2 induced decidualization defects in stromal cells, which were specifically manifested as abnormal expression of IGFBP1, PRL, and FOXO1, suggesting that G0S2 may be involved in the regulation of decidualization and induction of decidualization defects. Moreover, in the rescue

experiment, co-knockdown of JAZF1 and G0S2 partially restored the high level of IGFBP1, but had no effect on PRL and FOXO1 expression, indicating that the JAZF1-G0S2 network is involved in regulating the expression of IGFBP1 but not the regulatory pathway of PRL and FOXO1. We have added this clarification to the manuscript, line:378-390.

(3) Line 407: I missed the part of the manuscript in which the decidualization defect was observed in RSA decidua. I suggest the authors include the data mentioned in line 422; otherwise, there is no clear association between decidualization defect and RSA in terms of the present study.

Response: The critical comment of the reviewer is highly appreciated. We apologize for not making this clear. The results of decidualization defect observed in RSA decidua have been requoted from our published data¹⁶. The results are as follows.

Fig. 3. Decidualization markers were also increased in RPL decidua tissues. (A-B) IGFBP1 protein level in RPL decidua tissues and control samples by IHC via tissue microarray (A) (n = 9) and western blot (B) (n = 20); (C-D) IGFBP1 mRNA level in RPL decidua tissues and control samples by in situ hybridization(C) and ScRNA-seq analysis in DS1 cluster; (E-F) qRT-PCR(E) and ELISA detection(F) showed the PRL expression level in RPL decidua tissues and control samples (E: control=23, RPL=22; F: control=11, RPL=12). ** indicates $P < 0.01$. **** indicates $P < 0.0001$.

(4) Line 423: the results were not confirmed in primary cell cultures.

Response: We sincerely appreciate the reviewer's important comment. We have replicated the results of the immortalized cell lines using the primary endometrial stromal cells, and the results were consistent (line: 172-176).

(5) Lines 451-457: This information summarizes the results. Many points are missing in the discussion.

Response: We appreciate the reviewer's rigorous comments. We rephrased the discussion as "Our results proved that JAZF1 worked as a typical transcription repressor just like in other systems. However, JAZF1 targeted apoptosis-related gene

G0S2 in immortalized HESCs during decidualization, implying cell type specific functions of JAZF1. Moreover, as a transcription repressor, whether JAZF1 directly regulates the transcription of key decidualization-related genes requires further experiments.” In addition, we have added the role of G0S2 and Pur β in decidualization in the discussion section, listing as “line:378-390. Previous studies have found that G0S2 induces cell apoptosis by binding to BCL2 and activating mitochondrial apoptosis pathway. In the current study, we found that G0S2 was a novel molecule that regulated stromal cells survival and decidualization. Mechanistic studies led to the discovery that JAZF1 depletion induced cell death via mitochondrial apoptosis in stromal cells through activation of G0S2. However, when JAZF1 and G0S2 were co-downregulated, they only down-regulated IGFBP1 expression, but not PRL expression, which suggests that PRL may be regulated by other downstream networks of JAZF1. Transcription factor Pur β has been found to bind to DNA promoter sequences of several genes and thus participate in transcriptional regulation, but the effect of its expression on cell function has not been described. This study showed that Pur β could bind to G0S2 promoter and promote its transcription, thus inducing cell death and decidualization defects. More importantly, knockdown of Pur β or G0S2 restored the apoptotic defects of JAZF1-knockdown HESCs or primary ESCs”.

METHODS

(1) Please include exclusion and inclusion criteria.

Response: Thanks for the comments. We have provided the exclusion and inclusion

criteria as following in the part of “Sample of clinical cases” in methods. The methods is shown as “line:417-418. Patients with endocrine disorders, metabolic, or autoimmune diseases were excluded. Only patients with unexplained RSA were recruited in the study”.

(2) STATISTICS: The authors stated they used statistical analysis to compare two variables; however, there are results with more than two variables.

Response: We sincerely appreciate the reviewer’s important comment. Line:619-621. Statistical analyses were performed using two-tailed unpaired Student’s t-test or Mann-Whitney U-test for comparisons between two groups and one-way analysis of variance (ANOVA) for comparisons among more than two groups. We have revised our text to clarify the statistical methods used in the study.

References

- 1 Vashistha, A., Khan, H. R. & Rudraiah, M. Role of cAMP/PKA/CREB pathway and β -arrestin 1 in LH induced luteolysis in pregnant rats. *Reproduction* **162**, 21-31 (2021). <https://doi.org:10.1530/REP-20-0661>
- 2 Huang, F. *et al.* FoxO1-mediated inhibition of STAT1 alleviates tubulointerstitial fibrosis and tubule apoptosis in diabetic kidney disease. *EBioMedicine* **48**, 491-504 (2019). <https://doi.org:10.1016/j.ebiom.2019.09.002>
- 3 Shang, J., Gao, Z.-Y., Zhang, L.-Y. & Wang, C.-Y. Over-expression of JAZF1 promotes cardiac microvascular endothelial cell proliferation and angiogenesis via activation of the Akt signaling pathway in rats with myocardial ischemia-reperfusion. *Cell Cycle* **18**, 1619-1634 (2019). <https://doi.org:10.1080/15384101.2019.1629774>
- 4 Zhou, Q. *et al.* EHD1 impairs decidualization by regulating the Wnt4/ β -catenin signaling pathway in recurrent implantation failure. *EBioMedicine* **50**, 343-354 (2019). <https://doi.org:10.1016/j.ebiom.2019.10.018>
- 5 Wang, Z. *et al.* ATF3 deficiency impairs the proliferative-secretory phase transition and decidualization in RIF patients. *Cell Death Dis* **12**, 387 (2021).

- <https://doi.org:10.1038/s41419-021-03679-8>
- 6 Rawlings, T. M. *et al.* Modelling the impact of decidual senescence on embryo implantation in human endometrial assembloids. *Elife* **10** (2021). <https://doi.org:10.7554/eLife.69603>
- 7 Yao, S. *et al.* Resveratrol alleviates zea-induced decidualization disturbance in human endometrial stromal cells. *Ecotoxicol Environ Saf* **207**, 111511 (2021). <https://doi.org:10.1016/j.ecoenv.2020.111511>
- 8 Lucas, E. S. *et al.* Recurrent pregnancy loss is associated with a pro-senescent decidual response during the peri-implantation window. *Commun Biol* **3**, 37 (2020). <https://doi.org:10.1038/s42003-020-0763-1>
- 9 Gellersen, B. & Brosens, J. J. Cyclic decidualization of the human endometrium in reproductive health and failure. *Endocr Rev* **35**, 851–905 (2014). <https://doi.org:10.1210/er.2014-1045>
- 10 Cha, J., Sun, X. & Dey, S. K. Mechanisms of implantation: strategies for successful pregnancy. *Nat Med* **18**, 1754–1767 (2012). <https://doi.org:10.1038/nm.3012>
- 11 Shao, J. *et al.* Estrogen promotes the growth of decidual stromal cells in human early pregnancy. *Mol Hum Reprod* **19**, 655–664 (2013). <https://doi.org:10.1093/molehr/gat034>
- 12 Altmäe, S. *et al.* Meta-signature of human endometrial receptivity: a meta-analysis and validation study of transcriptomic biomarkers. *Sci Rep* **7**, 10077 (2017). <https://doi.org:10.1038/s41598-017-10098-3>
- 13 Berkhout, R. P., Lambalk, C. B., Repping, S., Hamer, G. & Mastenbroek, S. Premature expression of the decidualization marker prolactin is associated with repeated implantation failure. *Gynecol Endocrinol* **36**, 360–364 (2020). <https://doi.org:10.1080/09513590.2019.1650344>
- 14 Iancu, M. E., Albu, A. I. & Albu, D. N. Prolactin Relationship with Fertility and In Vitro Fertilization Outcomes—A Review of the Literature. *Pharmaceuticals (Basel)* **16** (2023). <https://doi.org:10.3390/ph16010122>
- 15 Panzan, M. Q. *et al.* Metoclopramide-induced hyperprolactinaemia caused marked decline in pinopodes and pregnancy rates in mice. *Hum Reprod* **21**, 2514–2520 (2006).
- 16 Zeng, S. *et al.* TNF α /TNFR1 signal induces excessive senescence of decidual stromal cells in recurrent pregnancy loss. *J Reprod Immunol* **155**, 103776 (2023). <https://doi.org:10.1016/j.jri.2022.103776>